# Temporal structure of mouse courtship vocalizations facilitates syllable labeling

Stav Hertz[1], Benjamin Weiner[2], Nisim Perets[1] & Michael London [1,2 ✉]

Mice emit sequences of ultrasonic vocalizations (USVs) but little is known about the rules governing their temporal order and no consensus exists on the classification of USVs into syllables. To address these questions, we recorded USVs during male-female courtship and found a significant temporal structure. We labeled USVs using three popular algorithms and found that there was no one-to-one relationships between their labels. As label assignment affects the high order temporal structure, we developed the Syntax Information Score (based on information theory) to rank labeling algorithms based on how well they predict the next syllable in a sequence. Finally, we derived a novel algorithm (Syntax Information Maximization) that utilizes sequence statistics to improve the clustering of individual USVs with respect to the underlying sequence structure. Improvement in USV classification is crucial for understanding neural control of vocalization. We demonstrate that USV syntax holds valuable information towards achieving this goal.

[1] The Alexander Silberman Institute of Life Sciences, The Hebrew University of Jerusalem, Jerusalem, Israel. [2] Edmond and Lily Safra Center for Brain Sciences, The Hebrew University of Jerusalem, Jerusalem, Israel. ✉email: mickey.london@mail.huji.ac.il

Mice emit ultrasonic vocalizations (USVs) in various behavioral contexts[1–8]. Recently, this behavior has gained interest as a proxy model for speech and language[9–13] and as a tool for behavioral phenotyping of neurodevelopmental disorders[14–18]. However, to fully gain access to the advantages provided by the information hidden in USVs, we need adequate methods to analyze this complex signal.

Observing the spectrogram of the sound signal (Fig. 1a), it is easy to appreciate that it is composed of distinct individual syllables where the power in the ultrasonic range (>20 kHz) is made of continuous stretches of positive power (USVs) or zero power (silence). The periods of silence between USVs (inter-syllable intervals, ISIs) follow a typical distribution with several distinct peaks, suggesting a prototypical process of producing these sounds. Thus, it is possible to parse the acoustic signal into individual USVs and USV sequences based on the distribution of ISIs (Fig. 1, and in refs. [3,12,19]).

A closer look at these individual vocalizations suggests the existence of classes. Many USVs have a rather simple form, composed mainly of narrow-band frequency sweeps. Some other USVs are composed of substructures of various sweeps, each of which can change its length and central frequency, resulting in a broad spectrum of shapes (Fig. 1a). Therefore, while the process of parsing the sound into individual USVs is primarily one of overcoming technical obstacles, the classification of the individual USVs into syllable classes based on their acoustic features presents complex and fundamental challenges.

Individual vocalizations of human speech can be assigned into well-known and distinct syllable classes. In sharp contrast, the existence, number, and identity of these classes are unknown when analyzing mouse USVs. Therefore, various approaches have been taken in developing methods of labeling syllables. For example, Holy and Guo[3] and later[19,20] have used frequency jumps as the main feature of differentiating individual USVs, and have labeled them according to the number and direction of the jumps while ignoring other features (e.g., duration). Alternatively, other algorithms have taken unsupervised learning approaches without deciding upfront on hardwired features[21–24]. When the same ensemble of USVs is labeled by different algorithms, there might not be a one-to-one mapping between the resulting labeling (Fig. 1b) and this will have important consequences, as discussed below.

Figure 1a highlights another property of mouse vocalizations, the existence of complex syllable sequences. Vocal communication systems in other species (e.g., human speech and bird songs) are also based on sequences of sound units. Songbirds are known to produce diverse and complex sequences or "songs"[25–27]. Some of these songs contain hierarchical acoustical units: notes, syllables, and motifs[28], which can be very stereotypic (e.g., Zebra finch[29,30]) or have an underlying complex syntax (e.g., Bengalese finches[31,32]). USVs produced by male mice during courtship share some of the syntax characteristics of bird songs[3,19,33].

Taken together, the nonhomologous assignment of labels by different labeling methods and the existence of complex structures in USVs sequences may lead to very different statistical properties of labeled sequences from the same data. For example, the distribution of the number of syllables with each label could take many forms, and similarly, the probability of a syllable pair appearing together in a sequence may vary. Therefore, the selection of the labeling algorithm may lead to different scientific conclusions and interpretations.

Here, we show that this undesirable consequence of having different syntaxes from the same data could, in fact, be useful in selecting and improving labeling methods. This is based on the observation that a syntax imposed by an algorithm dictates how well it predicts the label of the next syllable in a sequence. A "meaningful" labeling should have a high predictive power, which could then be taken as a measure for the labeling quality. Moreover, the information that is present in the sequence structure could improve the labeling. Using information theory tools, we examine these ideas by comparing the predictive power of various labeling methods, and we suggest a simple way to incorporate optimization of the predictive power as an integral driving force of a labeling algorithm.

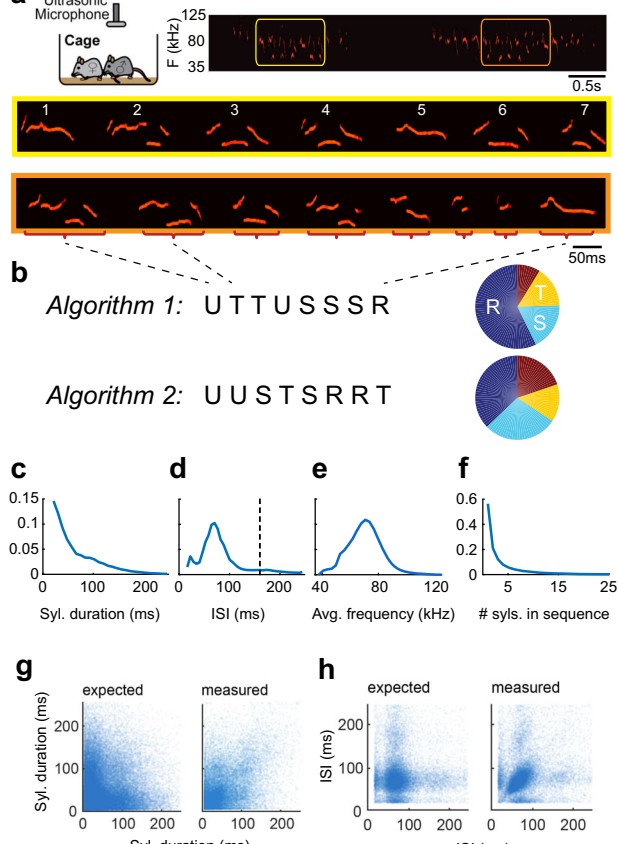

**Fig. 1 Parsing and labeling of USV sequences. a** A schematic diagram of the recording setup and an example spectrogram of USV recording showing two sequences. The yellow and orange frames zooming in on example USV sequences composed of several syllables, displaying the richness of shapes. Some shapes have distinctive features that may lead to natural categorization, such as sudden jumps in frequency (e.g., syl. #3, #6). However, syllables differ in many ways, including the duration of subcomponents (e.g., 3, 6, 7) or the number of sub-components (e.g., 1, 2 or 3, 4). Other features, e.g., syllable duration, have a continuous distribution over a wide range, and their use for the distinction between syllable types is less obvious (see also panel (**c**)). **b** Labeling examples by two illustrative algorithms. Syllables are labeled with four labels, marked by the letters: "RSTU". Algorithm 1 labels by the number of subcomponents while Algorithm 2 by syllable duration. The distributions of the labels, presented in the pie charts, indicate that the algorithms are not homologous. **c–f** Distributions of syllables parameters collected from ~345,000 syllables and ~33,000 sequences. **c** Syllable duration, **d** inter-syllable interval (ISI), **e** average syllable frequency, and **f** number of syllables in a sequence. **g** Correlation pattern between the durations of adjacent syllables in sequences: expected pattern based on (**c**), and measured pattern in the data). **h** same as **g** but for ISIs.

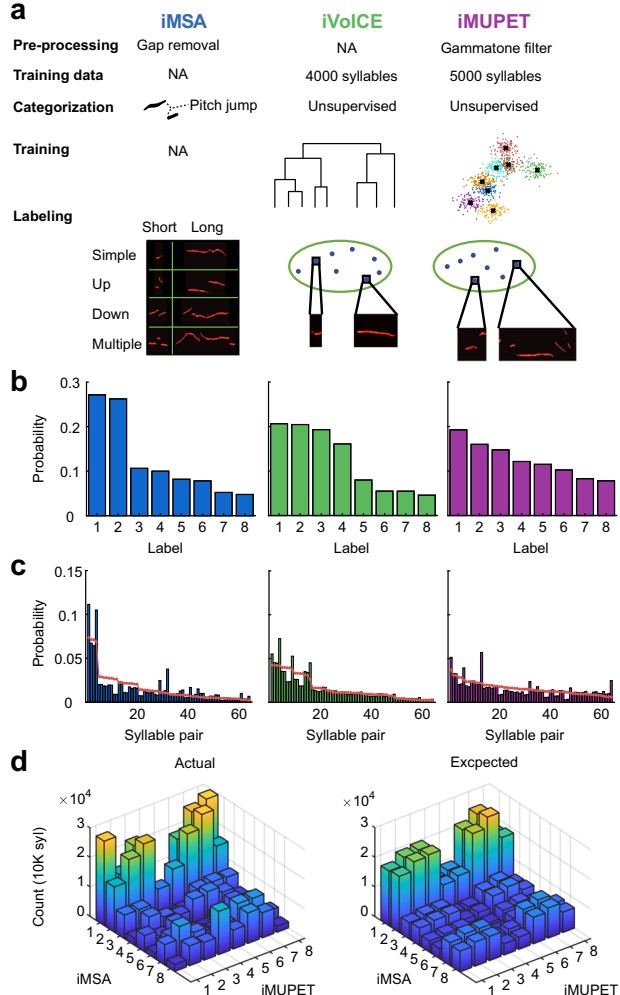

**Fig. 2 Labeling of the same data with different algorithms. a** A concise representation of the different algorithms that were adapted for this study. iMSA (left column) first preprocesses the data for gap removal. It requires no training data and labels the syllables based on their pitch jumps. The four basic pitch jump labels: Simple (no pitch jump), Up, Down, and Multiple. Each is then divided into two according to its median syllable duration for a total of eight labels. iVoICE performs hierarchical clustering on a training subset of 4000 syllables resulting in eight centroids that are then used to label the rest of the syllables based on a similarity measure. iMUPET algorithm performs a preprocessing gammatone filter on all syllables and then uses the k-means algorithm to create centroids from 5000 syllables. These centroids are used to label the rest of the syllables based on the cosine distance between the filtered syllable and the centroid. **b** The distributions of the labeled USVs from the database are shown for each algorithm, iMSA produces the most nonuniform distribution, and iMUPET the most uniform one. The difference between the distributions means that the algorithms are nonhomologous. **c** The distributions for pairs of USVs are shown for each algorithm. The red line depicts the predicted distributions derived from (**b**) assuming independence. The histograms are sorted by the expected distribution. **d** Actual joint distribution of the labels of iMSA and iMUPET for all USVs in the dataset and the expected joint distribution assuming independence.

## Results

**Analysis of basic USV properties suggests high-order structures.** To study the differences between labeling algorithms we compiled a database from our USV recordings. The recordings were made during sessions of interaction between adult male and female mice for a total of 78 h. We developed an analysis toolkit

(publicly available, see Methods) with a parsing algorithm to detect in the audio files the exact start and end times of each USV and each sequence of USVs. Applying this algorithm to our recordings, we extracted 346,632 USVs, which were then stored in the database along with their features. The individual USVs were grouped into 33,481 sequences.

Figure 1c–f shows three fundamental properties of USVs in our database: syllable duration, ISI, and syllable mean frequency. The occurrence frequency of syllable duration fits a monotonically decreasing exponential (two-sample Kolmogorov–Smirnov test: $D(1000) = 0.036$, $p > 0.05$ ($p = 0.56$); see Methods for more details) (Fig. 1c). The distribution of ISIs had two peaks, in agreement with previous reports[12,19]; however, in our hands, the two peaks occurred at shorter durations than previously reported. The first peak was at 20 ms and a larger peak at around 70 ms. Careful observation of the ISI distribution of different individual mice revealed a more complex picture, in which some mice had these double-peak distributions while others did not (Supplementary Fig. 1). Figure 1f shows the sequence length distribution in the database. Shorter sequences are more common than longer ones, and the distribution has a one-term decreasing power series fit (two-sample Kolmogorov–Smirnov test: $D(1000) = 0.026$, $p > 0.05$ ($p = 0.78$)).

We found a significant correlation in the distribution of syllable duration (Pearson's $r$ test: $r = 0.44$, $p < 0.001$), such that short syllables tend to follow short syllables and long ones to follow long ones (Fig. 1g). Similar results were found for the correlation of ISIs (Fig. 1h, Pearson's $r$ test: $r = 0.17$, $p < 0.001$). In conclusion, the existence of correlations already at this level of analysis (i.e., before labeling) suggests that USVs are not emitted independently of each other and that USV sequences have a nonobvious temporal structure.

**Labeling of the same USVs with different algorithms**. To test if different labeling methods are homologous, we chose three labeling algorithms that were recently published: MSA v1.3[19], VoICE[21], and MUPET[22] (Fig. 2). We chose these algorithms because (1) they represent different approaches to labeling, (2) they require relatively low manual involvement, and (3) the published algorithm provided code that could be applied to our database with relatively minor modifications (see Methods). Here, we refer to them as iMSA, iVoICE, and iMUPET to emphasize that we used the modified algorithm, which was inspired by the original one.

Figure 2a describes the key properties and the workflow of the three methods. In short, iMSA is based on hardwired features (pitch jumps), while iVoICE and iMUPET apply unsupervised learning to cluster syllables. This clustering is done on a learning set of a few thousands of USVs, resulting in a set of centroids which represent the different clusters. iVoICE uses an hierarchical clustering strategy, and iMUPET uses the k-means clustering algorithm[34] with a user-chosen predefined number of clusters. The label of a USV is obtained by assigning each observed USV to the closest cluster representative (centroid) using a similarity metric function that is specific for each algorithm (spectral similarity in the case of iVoICE and cosine distance in iMUPET). Two example centroids are shown for each of the algorithms.

To compare the algorithms, we ran all of them with eight labels. This gave a good balance of rich labeling on the one hand, while still enabling the collection of enough higher-order statistics, which will be important for later analysis. The number of labels is a natural parameter for the iVoICE and iMUPET algorithms, however, iMSA assumes only four labels, so we obtained eight labels by splitting each label into two according to the median syllable duration (Methods). Supplementary Figure 2

presents example syllables that were sampled from each one of the eight labels created by the three algorithms.

Figure 2b shows the distribution of the number of syllables assigned to each of the eight labels for the three algorithms. Because iMSA labels all syllables with no pitch jump as Simple (long or short), the first two labels occupy over 50% of the data and create a very non-uniform distribution compared to the two other algorithms.

Like the analysis in Fig. 1g, we also computed the distributions of pairs of labeled syllables imposed by the three algorithms (Fig. 2c). The red line represents the expected distribution (assuming statistical independence) derived from the distributions in Fig. 2b (this distribution was used for sorting the histograms). It is easy to see that in all cases, there are deviations from the expected distribution, implying that all algorithms capture some high-order structure of USV sequences. However, it is less obvious to deduce from that which algorithm captures more of this complexity.

Even without comparing the labels of individual USVs, the differences in the distributions in Fig. 2b already show that the algorithms label USVs in a nonhomologous manner (i.e., it is not that the labels can be permuted to obtain similar labeling) as suggested in the introduction. However, it is possible that they agree on the majority of the USVs, and there is a relatively small group of USVs that are labeled differently. To better test this possibility, we constructed the joint distribution between two algorithms. For each USV in the database, we looked at the pair of its label assigned by iMSA and iMUPET, and counted USVs for each of these pairs. If this option was true, one would expect that in each row in the joint matrix there would be one column with a significantly high count, however, as depicted in Fig. 2f, this is not the case. For most labels of iMSA, there is a fairly uniform distribution of the count over the iMUPET labels and vice versa. This distribution further strengthens the conclusion that the algorithms are not homologous. On the other hand, comparing this joint distribution with the expected one (Fig. 2d) reveals that the mapping is also not independent. Similarly, Supplementary Fig. 3 shows that when comparing the assignments of iMSA and iVoICE there is also no one-to-one mapping between the assigned labels.

**The predictive power of labeling algorithms**. Figure 2c suggests that the difference between the distributions (of pairs of labels) imposed by the algorithms on the same data may be used to quantify the differences between them. Based on this observation, we propose a framework for evaluating labeling algorithms. The guiding principle is that a labeling that exposes regularities in vocalization sequences is more likely to capture their underlying statistical structure. The better the statistical model of the USV sequences, the better the prediction it allows to draw about the future of the sequence.

Therefore, we evaluate an algorithm by quantifying how well the syntax model it imposes predicts the future of the sequence.

Given a USV dataset and a labeling method, the quantification process is done in two steps: (1) generating the syntax model and (2) calculating the model's predictability. For step 1 (syntax model), we applied a labeling algorithm to the USVs in the database and obtained sequences of labeled USVs. We used Markov chains of different order to represent the syntax model and to account for the dependence of a syllable on its prefix. Given that we assume that USVs come as individual syllables and are assigned one out of a given number of labels, we consider only discrete time and state-space Markov chains. In order to estimate the model parameters from the data, we assumed that the underlying Markov process is stationary and irreducible[35]. In our

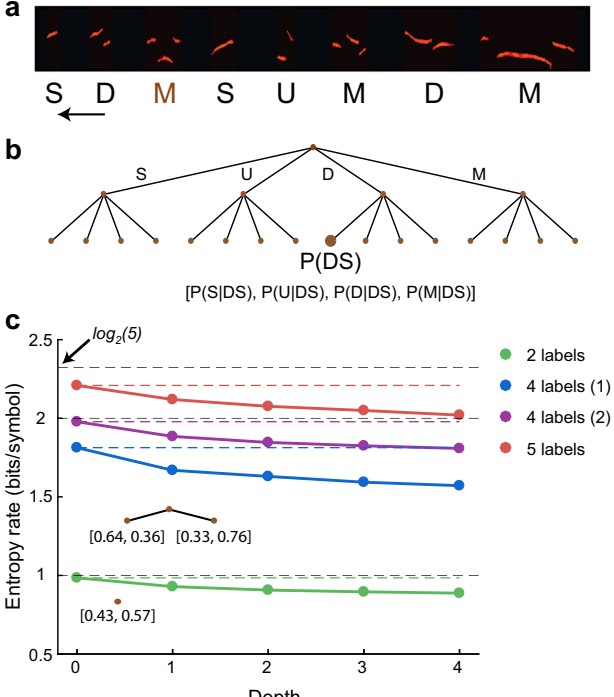

**Fig. 3 Computing the entropy rate of a labeling method. a** The labeling algorithm is applied to every given sequence (in this example iMSA with four labels). **b** A suffix tree is constructed to hold the counts for a label given the suffix preceding it. Here, an example for a tree of depth $D = 2$, the label $M$ is preceded by the suffix DS (read from right to left) and thus the leaf $M|DS$ counter is increased by 1 and the probability $P(M|DS)$ is updated. This constructs a distribution from which the entropy rate of the syntax model is calculated. **c** Entropy rate values of four different labeling algorithms (each marked by a different color; see text). The x-axis represents the depth of the tree, the solid lines represent the entropy rate in each depth and the dotted colored lines represent the entropy rate of the 0th-order model. The dotted black lines mark the upper bound on the entropy rate for a given number of labels $N_c$ (calculated as $\log_2(N_c)$). Inset trees represent the suffix trees with depth 1 and 2 for the algorithm that has two labels (green).

case, this means that (1) we assume that the underlying probabilities of label sequences are not changing with time (or between recording sessions), and (2) it is possible to get to any state from any other state in the Markov chain. We represent the $m$th-order Markov model as a Suffix tree of depth $m$ (Fig. 3A–C[36]). Note that in our notation, a suffix is read backwards in time, from right to left. In the leaves of the suffix tree we store (1) the probability to obtain the suffix represented by the branch leading to the leaf. For example, for the leaf in Fig. 3 (e.g., in the branch DS, we keep $p(DS)$ which counts how many times the pair SD has appeared in all the sequences relative to all pairs in all sequences. (2) The conditional probabilities for each label given the suffix leading to the leaf. For example, $p(M|DS)$, which is how many times $M$ appeared following the pair SD, relative to all the appearances of the pair SD.

In step 2, for a given suffix tree we evaluate its predictive power by calculating the entropy rate of the $m$th-order Markov model. The entropy rate quantifies the amount of uncertainty regarding the label of the next USV given the syntax model and the labeled syllables in the suffix of length $m$. A low entropy rate (bounded by 0 bits/symbol from below) means a high amount of predictability while a high entropy rate means a high degree of uncertainty (and is bounded from above by the log of the number of possible

labels). The entropy rate is given by $H_m = -\sum_{ij} \mu_i P_{ij} \log P_{ij}$ where $\mu_i$ represents the probability of obtaining the suffix represented by the branch leading to the $i$th leaf and $P_{ij}$ represents the conditional probability of the $j$th label given the suffix represented by the $i$th leaf[35].

**Definition of the SIS**. Figure 3c demonstrates the application of this strategy for a few illustrative cases. We labeled our USV database using four simple labeling algorithms, and for each algorithm computed the entropy rate for suffix tree models as a function of the tree depths. The first algorithm (depicted in green) uses only two labels: USVs containing pitch jump (J) or no pitch jump (N). The entropy rate of this model is bounded from above by 1 bit/symbol (dashed blue curve; when N and J appear independently and with equal probability). The distribution of the J and N syllables in our data was (43, 57%) and therefore with the 0th-order Markov model the entropy rate is 0.98 bits/symbol (Fig. 3c). For a 1st-order Markov model, the entropy rate decreases to 0.93 bit/symbol due to the apparent tendency of J syllables to follow J (64%) and N to follow N syllables (67%). Computing the entropy rate for higher-order Markov models shows a modest decrease with the order, which saturates at order 4 (i.e., the additional contribution of the syllables beyond the first 4 in the suffix is negligible for predicting the label of the next syllable).

Next, we considered two additional labeling algorithms. The first is iMSA with the four original labels (Simple, Up, Down, and Multiple; depicted in blue). The second was a variant where the Up and Down labels were merged, and the Simple label was split into Short and Long syllables (depicted in purple). With four labels, the entropy rate is upper-bounded by 2 bits/symbol and for the 0th-order Markov model both schemes are slightly below that bound because the distribution of labels is not uniform. Moreover, their 0th-order estimate of the entropy rate is not equal. Like in the previous case, the entropy rate of both algorithms decreases with increasing order of the Markov model. Note that in all the cases, the entropy rate is significantly larger than 0, implying that the production of a syllable is probabilistic and even given the full history of the sequence, uncertainty remains regarding its label. Lastly, it is worth noting (and will be discussed further below) that the amount of drop in the entropy rate between the 0th-order model (1.81 and 1.98, respectively) and the 4th-order model (1.57 and 1.8, respectively) is not identical. Hence, the information gain between the case where the previous syllable is unknown and the case where we know the previous four syllables is not the same for the two models. We also considered an algorithm with five labels (Simple-Short, Simple-Long, Up, Down, Multiple; depicted in red), which strengthens our conclusions from the previous models, namely that the 0th-order entropy rate is close to $\log_2(5)$ bits/symbol and that the reduction in entropy rate saturates at 4th-order Markov model.

The analysis presented in Fig. 3c shows that the entropy rate could be useful for comparing various labeling algorithms. It also, however, highlights several subtleties. Firstly, the more labels the algorithm assigns, the more likely its entropy rate will increase. This happens because the upper bound on the entropy increases logarithmically with the number of labels (as the number of possible labels increases, so does the uncertainty regarding the label of the upcoming syllable). A meaningful comparison using entropy rate is possible only if the two algorithms use the same number of labels (or, alternatively, one could normalize by the upper bound). A second issue is that using entropy rate as a predictive measure does not distinguish between data lacking temporal structure and a poorly performing algorithm (an

algorithm that assigns random labels to USVs). Indeed, using syntax to compare labeling algorithms is only meaningful when the data itself has structure. In our case it is evident that USV sequences have high order temporal structure (Fig. 2c).

Lastly, the examples in Fig. 3c highlight that an algorithm can achieve high predictability (low entropy rate) simply by assigning the same label to every USV independent of any acoustic feature. In such a case, we know with certainty what will be the label that the algorithm will assign for the upcoming syllable because it is always the same one. Unfortunately, this is the exact opposite result of finding regularities in the data. Rather, it is imposing "fake" regularities by the algorithm. We conclude that merely using entropy rate as a measure for comparing algorithms is not ideal because the more nonuniform the distribution of labels at the 0th-order is, the lower the entropy rate, and the higher the predictability, irrespective of the true complexity of the sequences.

To overcome these challenges, we note that the entropy rate of the 0th-order distribution is an inherent property of the labeling algorithm. Since labeling algorithms consider one USV at a time (independently of the order of which they appear in the sequence), it is unlikely that they introduce regularities of high order beyond those they impose on the 0th-order distribution. Therefore, a measure that is insensitive to the 0th-order distribution is more suitable for our purposes.

We claim that for sequences of labeled syllables $X_{n-D}, \ldots, X_n$ the mutual information (MI) between the suffix and the next syllable $I(X_n ; X_{n-1}, \ldots, X_{n-D})$ provides a better measure to quantify how knowledge of recent syllables in a sequence affects our prediction of the next syllable. We denote this specific MI as syntax information score (SIS; measured in units of bits/symbol, see Methods). In our case the MI is equal to $H(X_n) - H(X_n|X_{n-1}, \ldots, X_{n-D})$[35], i.e., how much our uncertainty regarding the next syllable drops when we are given the prefix. Note that for a $D$-order Markov chain $H(X_n|X_{n-1}, \ldots, X_{n-D})$ is actually the entropy rate of the process[35] and therefore the MI is given by the difference between the entropy rate at order $D$ and at order 0 (see Fig. 3c).

The SIS being MI is bounded from above by the $H(X_n)$, which is the entropy of the 0th-order model. Consider the case of two labeling algorithms that use the same number of labels, where algorithm 1 imposes a very biased labeling (tending to assign almost all USVs the same label) and algorithm 2 results with a more balanced labeling at 0th-order. The entropy rate of the 0th-order for algorithm 1 will be smaller than that of algorithm 2, which sets an upper bound on the SIS as described above. Therefore, algorithm 2 has a larger range to find regularities in the higher-order distribution, while algorithm 1 is penalized for the highly nonuniform distribution it imposes on the 0th-order distribution. Indeed, at the limit, an algorithm that assigns all USVs with the same label will have 0 entropy for the 0th-order distribution, enforcing SIS of 0 bits/symbol for any higher-order model, and consequently the lowest predictability measure possible, in line with what we expect from our measure. On the other extreme, an algorithm that assigns random labels to USVs will achieve the highest possible entropy at 0th-order, but at the same time, this entropy will not decrease for higher orders, setting the SIS to 0 bits/symbol as well. Lastly, a dataset of sequences with no temporal structure will result in SIS of 0 bits/symbol independent of the algorithm being used. We, therefore, conclude that the SIS may serve as a good candidate for comparing labeling algorithms and will be used below. To validate our ability to estimate the true entropy rate and SIS values from the data, we ran simulations on synthetic Markov models for which these values can be computed analytically. Supplementary Figure 4 shows that our estimations converge to the analytical values in all cases tested.

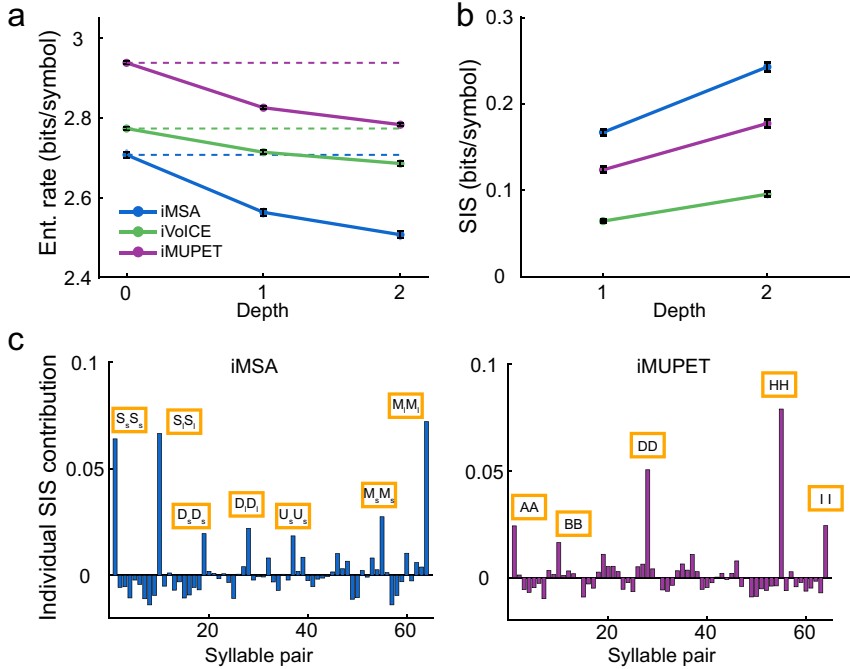

**Fig. 4 Comparison of the labeling methods. a** Entropy rate computed for the models produced by the three different methods (with 8 labels) for tree depth of order 0–2. Solid lines represent the entropy rate for each depth; dotted lines represent the rate for zero depth. The black error bars represent 2 s.d., computed over 25 repetitions. In each repetition, 60% of the sequences in the database were used to construct the suffix tree (see Methods). **b** The syntax information score computed for the models of depth 1 and 2 in bits/symbol. The values are calculated as the mutual information between the next syllable and its prefix. **c** The individual contribution of each pair to the total SIS value (depth 1).

**Comparing the SIS for existing labeling methods**. We measured the SIS for the three labeling algorithms presented in Fig. 2 applied to the USV database with eight labels. Figure 4a plots the entropy rates for the three algorithms for different depth of the suffix tree. Note that iMSA has the lowest 0th-order entropy as expected from Fig. 2b, while iMUPET, which has the most uniform distribution at 0th-order, yields an entropy rate of 2.9 bits/symbol (close to the maximum of 3 bits/symbol for 8 labels). However, it is easy to see the iMSA has the largest drop between the entropy rate at the 0th-order and the higher-order ones. Figure 4b shows the SIS for all three algorithms for $D = 1$ and $D = 2$. The graph shows that iMSA yields the highest SIS for both depths, despite its lower 0th-order entropy rate. iMUPET produces the second-best result. This result suggests that the frequency jumps in USVs are likely to be an important feature in their classification. We conclude that our framework and the SIS measure is a feasible method that is sensitive enough to measure differences in existing labeling methods and highlight the method that best captures regularities in the data.

The SIS increases for algorithms that detect regularities in the higher-order structure of the sequences. In order to see what these regularities are, we note that the SIS can be computed as a sum of the individual contributions of each n-tuple (see Methods). Therefore, we looked at the contribution of individual syllable pairs (of the possible 64 pairs). The contribution of a given observed pair $x_{n-1}, x_n$ is calculated as $P_i \log \frac{P_i}{Q_i}$ (equivalent to the KL-divergence between $P$ and $Q$) where $P_i = p(x_n, x_{n-1})$ and $Q_i = p(x_n) * p(x_{n-1})$ (i.e., the case where the next syllable is independent of the suffix). A similar analysis was carried out for triplets in Supplementary Fig. 5. Please see Methods for the formulation of the general case of $D$-order suffix. For a given pair, if $P_i = Q_i$ then the contribution of this pair to the SIS is zero. However, for some pairs this value could be significant. Figure 4c plots these values for both iMSA and iMUPET (see

Supplementary Fig. 5 for all three algorithms). It is clear that in both cases the pairs that obtain the highest values are repetitions of the same syllable (i.e., pairs of identical syllables). Interestingly, however, for iMSA, the "Simple" syllables, which in the original algorithm do not consider the duration of the syllable, show repetition only for syllable of similar duration. The pairs of Simple-long and Simple-short appear more than expected from their occurrence probabilities (Fig. 2b) assuming independence. However, pairs of Simple-long followed by Simple-short (or vice versa) actually show up less than expected. While splitting the Simple category into two labels by the median duration, as done here, is quite arbitrary, it may indicate that a group of the USVs labeled as Simple might be subdivided into subgroups (possibly such that the feature of syllable duration plays an important role in this division), and that finding these subgroups will reveal more of the richness of the statistical structure of USV sequences. This analysis indicates an interesting relationship between the SIS and the occurrence of unexpected motifs in the data (see Discussion).

In this analysis we have used eight labels for each algorithm. Some algorithms, such as MUPET, emphasize that a larger number of labels are required to achieve good clustering. Limiting the number of clusters may result in misclustering USVs, which may have a significant effect on the predictability of the next syllable, and therefore a detrimental effect on the SIS. On the other hand, increasing the number of clusters ($N_c$), as shown in Fig. 3, will result in an increase of the entropies (increase the uncertainty regarding the next syllable) and likely affect the SIS. In order to look into this effect in more details, we have calculated the SIS for the iMUPET algorithm for $N_c$ ranging from 8 to 64 for a tree of depth 1; and $N_c$ of 8 and 16 for a tree of depth 2 (Fig. 5, see also Supplementary Fig. 6). Note that for $N_c > 64$ for depth 1 and $N_c > 16$ for depth 2 we could not obtain a valid entropy estimation[37,38]. Clearly, the entropy of the 0th-order distribution

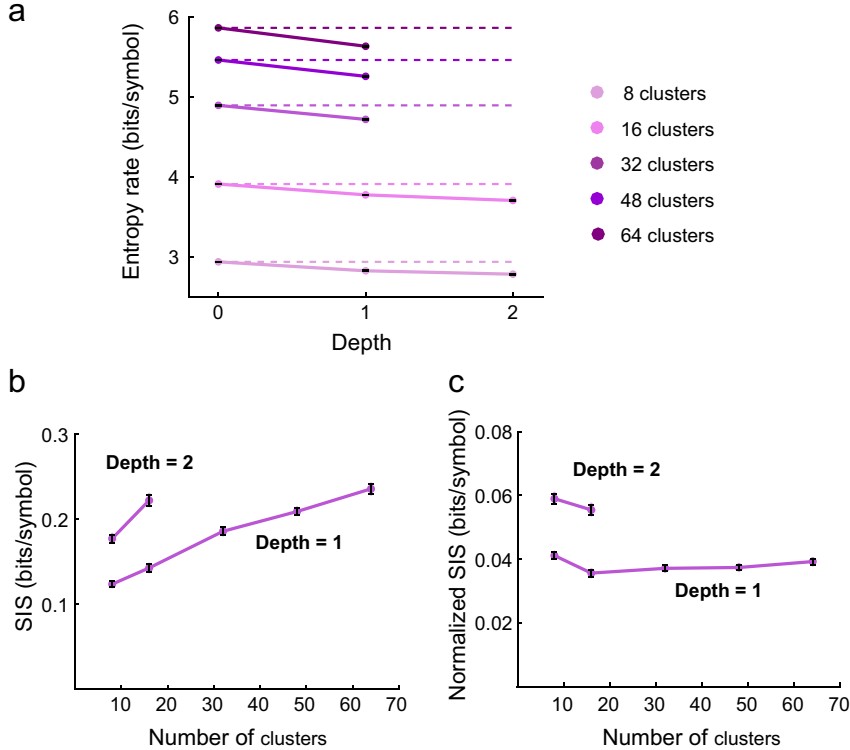

**Fig. 5 Effect of increasing the number of clusters for the iMUPET algorithm. a** Entropy rate values computed for the models produced by applying the iMUPET algorithm on the USV dataset with an increasing number of clusters. The criterion for estimating the entropy rate and SIS of a given suffix tree is that the amount of conditional probabilities with a 0 value (i.e., no occurrence of the equivalent $(D+1)$-tuple) is lower than 10%. **b** SIS values for these models computed for depth 1 and 2. **c** Normalized SIS values for the models. The Normalization factor is equal to $\log2(N_c)$ where $N_c$ is the number of clusters used.

increases monotonically with an increase in $N_c$ (approximately as $\log_2(N_c)$) (Figs. 3c and 5a). Similarly, the entropy rate for higher-order models also increases with $N_c$ but it is not clear a priori what is the dependency of SIS on $N_c$. Figure 5b shows that in this case, the SIS does increase as a function of $N_c$ which means that increasing the number of clusters does improve the predictability of the next syllable given the suffix. Interestingly, when considering tree of depth 1, the point at which iMUPET obtains a larger SIS as compared to iMSA with 8 labels is $N_c = 32$, and for a tree of depth 2 we have not had enough data to find that point (compare Figs. 4b and 5b). Notice that the increase of SIS with $N_c$ comes at the price of an exponential increase in the complexity of the model. It is therefore interesting to consider how the increase in the SIS is compared to the number of bits required to encode each syllable. This is obtained by normalizing SIS by $\log_2(N_c)$ and as shown in Fig. 5c this normalized SIS does not have a clear dependency on the number of clusters (see Discussion).

**Improving the SIS of current algorithms.** Labeling algorithms are forced by their very nature to assign a single label to each USV, even in cases where the decision is not obvious. This is especially evident for clustering-based algorithms such as VoICE and MUPET. This difficulty arises from a combination of the lack of a natural measure of similarity between USVs and the lack of separability between clusters (often, the "clouds" around each centroid have overlapping volumes creating some level of ambiguity). This poses a substantial challenge when attempting to categorize syllables. Even when using "soft clustering"[39], where a probability of assigning a label to the USV is computed for all the labels, the algorithm would eventually be forced to assign the most likely label. The higher-order statistics that were showing up

in our previous analysis, suggest that the sequence data may hold information that can assist labeling in such case of ambiguity. This is analogous to trying to parse a note written with poor handwriting and deducing that a certain letter is likely to be a U rather than a V because it follows the letter Q. Figure 6a illustrates such an example, where the probability assigned by an algorithm for a USV to be labeled as S is larger than its probability to be assigned the label T. If we assign this USV the label S, this assignment, however, has also an effect on the syntax model. The next USV in the sequence will have a higher probability of following S and a lower probability to follow T. This effect can be evaluated using the SIS. Doing so may reveal that assignment of T would, in fact, increase the SIS more than the assignment of S. If this difference in the SIS in the two cases is large enough, we may decide that this USV should be labeled T after all, and ignore our feature-based similarity measure that is used by the labeling algorithm.

We present the syntax information maximization (SIM) algorithm that considers the SIS of the labeling as an optimization constraint. Given a set of centroids, the goal is to find a new set that has a larger SIS on a test set (that it was not trained upon). That means that it has to consider the properties of the single syllable as well as the syntax. To test this approach, it is useful to use an algorithm that is bounded by the SIS of the other algorithms, which give a natural scale to the comparison. For that reason, we have chosen to use iMUPET (which achieved the second-best SIS in our test, Fig. 4b) as a starting point for the SIM algorithm. Figure 6b illustrates the process. We choose a training set of USVs (composed of half of the sequences in the database) and use iMUPET to compute centroids that represent the different clusters. Next, each of the centroids is perturbed in

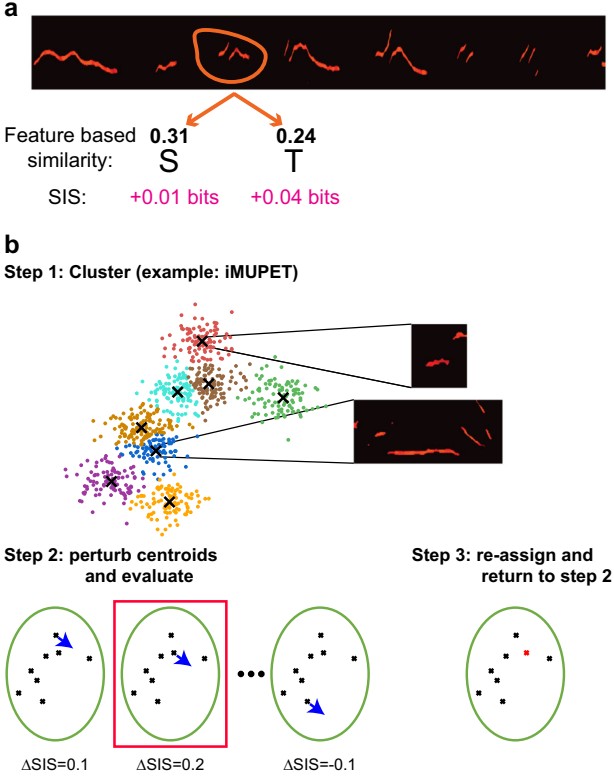

**a**

Feature based  **0.31**       **0.24**
similarity:        **S**           **T**

SIS:        +0.01 bits    +0.04 bits

**b**

**Step 1: Cluster (example: iMUPET)**

**Step 2: perturb centroids**          **Step 3: re-assign and**
**and evaluate**                        **return to step 2**

ΔSIS=0.1      ΔSIS=0.2      ΔSIS=-0.1

**Fig. 6 Syntax information maximization (SIM) algorithm. a** Labeling of a syllable is based on the feature similarity of the syllable to each cluster. The similarity defines the probability of the syllable to belong to each cluster. The syntax information score (SIS) of the labeling represents the amount of information that the labeling provides about the next syllable in a sequence. SIM optimizes clustering by using the SIS as an additional constraint. **b** Illustration of the SIM algorithm. The initial condition is the clustering of another algorithm. Here we chose iMUPET as an example. iMUPET provides a set of centroids and each USV is assigned to a centroid. In step 2 a random perturbation is chosen and each of the centroids is perturbed in turn. Then, the change in the SIS is evaluated (ΔSIS). The perturbation that resulted in the largest ΔSIS is chosen (red frame). In step 3 all the USVs are relabeled based on the new set of centroids (all but the chosen centroids are the same, and the chosen centroid is replaced by its perturbed version (red dot)). Step 2 and Step 3 are repeated until convergence is achieved.

turns, with an identical random perturbation. For every perturbation, all the USVs are re-assigned to clusters and the change in SIS (ΔSIS) is evaluated on the resulting syntax (still on the training set). After the ΔSIS has been evaluated for all the sets of perturbed centroids, the perturbation that resulted in the largest ΔSIS is chosen. The USVs are re-assigned and the procedure is repeated.

The results of the algorithm are shown in Fig. 7a. The algorithm is designed to increase the SIS on the training set in each step, and therefore it is not surprising that indeed the SIS is improving on this set. However, we also evaluated the algorithm on the test set after each step and as seen the SIS shows a similar trend on the test set that is not considered during the iteration of the algorithm. This demonstrates that the new set of centroids found by the algorithm generalizes well and captures better the syntax of the USV sequences. Note that for depth 1, the algorithm yields an SIS that is larger than that of iMSA. For depth 2, the algorithm obtains an SIS that is slightly lower than iMSA (but significantly larger than the initial point). Figure 7b plots 0th-order and 1st-order distributions for the resulting labels of SIM.

Notice that these distributions are different from those of the algorithms shown in Fig. 2b, c. Moreover, as seen in Fig. 6c, SIM captures pairs of syllables that have an unpredictable occurrence (~0.1). Lastly, Fig. 7d compares SIM to the other algorithms and shows that the new algorithm obtained improved SIS for both 1st-order and 2nd-order models over the original iMUPET algorithms and is comparable to that of the iMSA.

## Discussion

The analysis of mouse USVs led to the development of many methods that assign discrete labels to USV syllables[19,21,22]. These proposed labeling methods are based solely on the spectral representation of every single syllable. However, there is currently no ground-truth that can serve as a standard to evaluate the performance and accuracy of these methods and thus the selection of which method to use is somewhat arbitrary. We showed that analyzing the syntax imposed by a labeling algorithm could provide a good evaluation measure for its performance.

Our comparison revealed that while the algorithms differ in their predictability, they all capture some meaningful features in the temporal structure of USV sequences. Assuming there is a real natural and true classification of USVs, there might be various reasons why different labeling algorithms achieve different SIS. One option is that each algorithm uses only a subset of the features which are required for a true classification. For example, it might be that pitch jumps are an essential feature for USV classification, but Simple upward sweeps are inherently different from downward sweeps, a feature that is not considered by MSA, but might appear as an implicit feature for the other algorithms. A second possible reason is that the representation created by the geometric projection and the similarity metric used by clustering algorithms cause distinct clusters of USVs to become inseparable. For example, MUPET pre-processes the frequency representation of USVs before projecting them into high-dimensional space and uses a cosine-based norm to cluster them. We could now use the SIS and compare the resulting labels using other metric measures and different pre-processing procedures over the same set of data to test if this pair is optimal.

Another way to consider the problem of using syntax to assist the labeling is to model it as an information channel[40]. Here, the channel has a source given by a Markov chain with a finite alphabet. The Markov input is transmitted through the channel where the label can be corrupted (flipped) by the noise of the channel. In this setting the noise is shaped by the bias of each labeling algorithm (e.g., the specific similarity measure used by the algorithm cause a USV on the border between two clusters to be assigned to the wrong cluster). This setup is a particular case of Hidden Markov models (HMMs)[41], and therefore the combination of SIS and algorithms for estimating HMMs could prove beneficial in the future for improving USV labeling.

Our vocalization database provided enough data to validly estimate the high-order statistics of the labels. To achieve this large database we recorded in a male–female interaction setup during all phases of interaction, which provided many more vocalizations than elicited in other contexts (e.g., female urine). However, this may result in a low occurrence rate of female vocalizations, which are not controlled for[42]. In addition, sessions of male–female encounter follow a stereotypic pattern where specific mating behavior (e.g., approach, sniffing, mounting, and intromission) tend to appear at different phases of the session. Moreover, the probability distribution of syllables appearance also changes with these phases[8,43]. It is possible that the syntax structure of USVs could relate to the mating context. While we have not addressed this issue in the study, careful experimental design that keeps for each USV the behavioral context can unravel even finer structures of USV

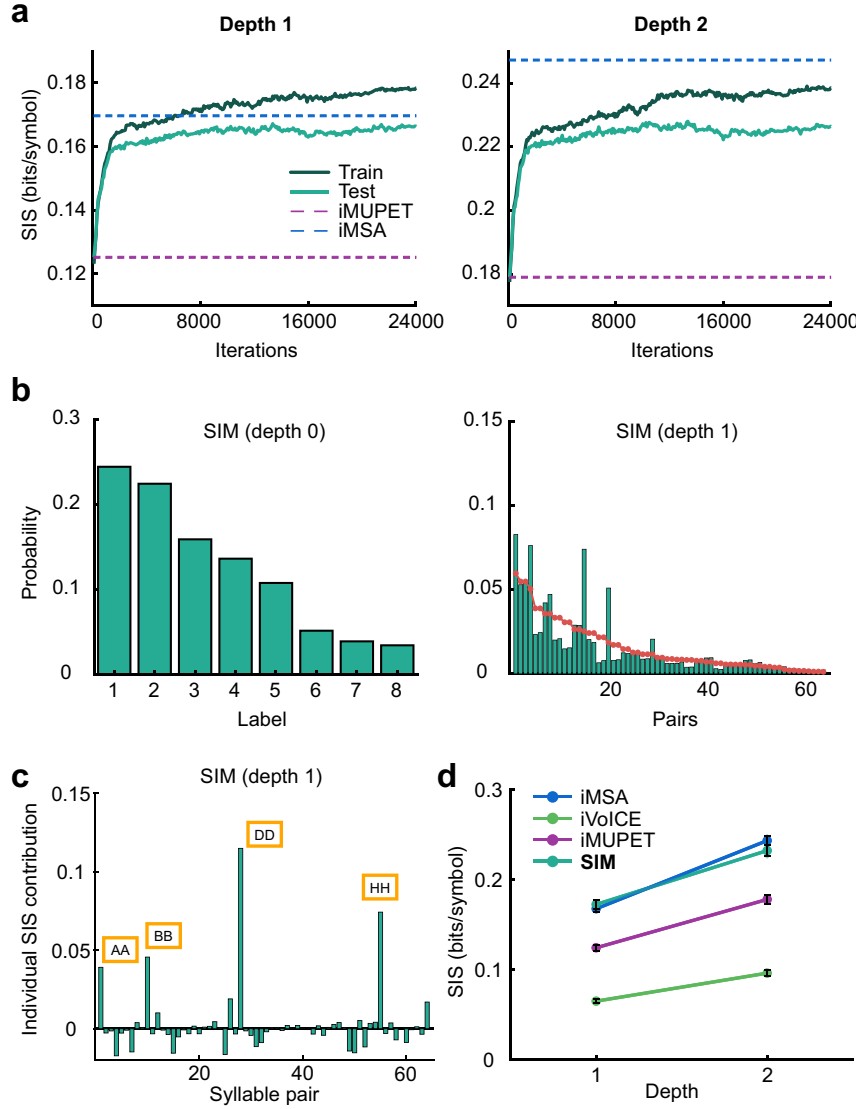

**Fig. 7 Results of the syntax information maximization (SIM). a** Results of the syntax information maximization algorithm from Fig. 6 on training and test sets with suffix trees of depth 1 and 2. The initial condition is the clustering of iMUPET (bottom dashed line). The algorithm quickly approaches the SIS of iMSA (top dashed line). **b** The distribution of labels and of syllable pairs of the SIM algorithm. **c** The contribution of each syllable pair to the SIS of depth 1. **d** The SIS for SIM compared to the original three algorithms for depth 1 and 2.

syntax, and make progress towards the goal of understanding the information USVs carry about behavior.

Previous studies have identified repeating motifs in sequences of USVs[3,44,45]. Our analysis addresses motifs in the labeled data, and emphasized "unexpected" motifs; namely when their probability for appearing deviates from the expected probability (given by the multiplication of the individual appearance probabilities of each syllable in the motif). Theoretically, our method could be applied for any motif length, but as the motif length increases, the number of sequences required to collect enough statistics increases exponentially. Therefore, we focused on motifs of length 2 and 3 (in Figs. 4c, 7c, and S5). This difficulty is also apparent in previous studies where statistical analyses of motifs were limited to motifs of order 3 or less[3,44]. While the SIS measure itself does not identify motifs, "unexpected" motifs contribute to an increase in the SIS. Therefore, if the SIS increases with the depth of the suffix tree, it is an indication that such "unexpected" motifs of that order exist. By examining the individual SIS contribution of each potential

motif, we were able to show that the most dominant ones included repeating syllables (Fig. 4c) for both the iMSA and iMUPET algorithms.

The SIM algorithm that we developed (Fig. 6) suggests that by considering the syntax when dividing the syllable space into clusters, the resulting predictability of the algorithm can be higher than the original one. Clearly, the algorithm we proposed is computationally suboptimal. The cost of each iteration, involving an evaluation of the syntax model at each step, is high. Nevertheless, our motivation was to demonstrate that the information existing in the syntax can be used to drive the algorithm to obtain an improved representation of the clusters, which generalizes to data that had not been considered before (the test set).

The exploration of the true number of classes in USV classification is not over yet. Most of our analysis was carried out assuming eight labels. This was done in order to balance between richness of labeling and availability of data that is required to construct valid high-order statistical models. Because some

methods suggest using a higher number of labels, we also explored a case with up to 64 labels. In our model, increasing $N_c$ increases the entropies (logarithmically). We found that increasing $N_c$ does increase the SIS. Our analysis shows that for a 1st-order Markov model, iMSA with 8 labels has larger SIS than iMUPET with 16 labels but lower than iMUPET with 32 labels. This implies that the initial gap between the two algorithms can be minimized, and even reversed, but at a cost of a fourfold increase in $N_c$. This improvement is of the same order of magnitude obtained by using SIM while keeping $N_c$ of 8, suggesting that combining SIM with iMUPET with increased $N_c$ may lead to even higher SIS than obtained in this study. Increasing $N_c$ also comes at a significant (exponential) increase in the complexity of the model (e.g., the number of states of the Markov model) and the amount of data that is required to establish a valid statistical model. To account for this increase, one can consider the normalization of the SIS by $\log_2(N_c)$. We found that after normalization the residual effect of increasing $N_c$ on the SIS is small. In conclusion, the balance between improved SIS and compact model is application dependent and can be tuned by the choice of $N_c$.

As we continue to develop labeling methods, the syntactic models they generate become more detailed and precise, enabling a better quantification of social disorders and their treatments. For example, we will be able to better measure the effect of a treatment or a specific gene knockout on ASD[12,46]. Therefore, carrying on the integration of such analytical tools together with behavioral paradigms could result in advanced treatments for social and speech disorders.

## Methods

**Animals**. For the recordings performed in our lab, we used C57BL/6 male and female mice (8–12 weeks old). All mice were group-housed (3–4 per cage) and kept on a 12 h light (7 a.m.–7 p.m.)/dark cycle with ad libitum food and water.

**Ethical note**. Experimental protocols were approved by the Hebrew University Animal Care and Use Committee and met guidelines of the National Institutes of Health Guide for the Care and Use of Laboratory Animals (IACUC NS-16-14216-3).

**USV database**. We created a database of 385 USV recording sessions of C57 male–female interactions. Most of the recordings were performed in our lab. For USV recording we used an UltraSoundGate system (Avisoft bioacoustic, Germany) composed of a CM16/CMPA ultrasound microphone, UltraSoundGate 116H computer interface, and USGH recorder software on a standard PC computer. A sampling frequency of 250 kHz and 16-bit recordings were used. For online monitoring we used simultaneous display of the spectrogram (256 points FFT). Thirty-six additional files were downloaded from the mouseTube[47] ("Female" context recordings from the "Social context comparisons" protocol[19]). All files were in "wav" format, and their length ranged between 2 and 30 min. Our approach in this work was to include different males, with different sexual experience and to search for the underlying common sequence structure. Therefore, many of the males were recorded several times and the sessions were scheduled independently of the females' estrous. The full set of files used in this study was uploaded to mouseTube and can be found using the group label "London Lab". In addition, Supplementary Table 1 contains information about the recording sessions and the male mouse that was recorded during that session.

**USV parser**. After testing a few parsing tools, we noticed that they were not optimized to cope with the different types of noise that existed in the USV files. These different levels of noise in the recordings were a result of: varying cage sizes, cage acoustics, locations of the recording device, and noise from the freely moving mice. Therefore, we developed a USV parser that is robust to these types of noise. Supplementary Figure 7 describes the flow of the parser. The parser receives as input one or more "wav" USV files and returns the start and end time of each syllable in the file.

**USV statistics**. The USV parser was applied to all recordings in the database. This resulted in 346,632 syllables. Using the start and end time of each syllable, we calculated the distributions of the syllable duration and ISI. We also examined the ISI distribution for several specific mice in order to test the variability between them.

In addition, the strongest frequency in each time point was detected and the mean frequency of each syllable was stored in the database, enabling the analysis of the mean frequency distribution. An ISI of more than 160 ms represented the end of the current sequence and the beginning of a new one. We calculated the number of syllables in the different sequences and created the sequence length distribution. To test the exponential fit of the duration probability density function (PDF) we performed a two-sided Kolmogorov–Smirnov test. This was done by calculating the fit parameters for the function $a*e^{b*x}$ (in our case, $a = 0.51$, $b = -0.41$). Then, we sampled 1000 values from the original and fit PDFs and ran the test on these values. The reported results are the average of 1000 repetitions of the sampling and testing process. A similar calculation was done for the sequence length, with the equation $a*x^b$ (with resulting parameters $a = 0.6$, $b = -1.88$).

For calculating the correlation between adjacent syllables, we collected all pairs of consecutive syllables in the sequences. We then ran a Pearson correlation test for both the duration and the ISI of the syllables.

**Adaptation of existing algorithms**. The source code for all three algorithms was available in MATLAB. We performed several adaptations to each algorithm in order to enable a fully automated execution.

*Mouse Song Analyzer v1.3*[19]: The MSA algorithm includes a built-in syllable parser. In order to label the same syllables for all algorithms, we replaced the syllables detected by the MSA parser with those that were detected by our parsing algorithm (see above). We ran the MSA algorithm on all files and saw that there were files where more than 5% of the syllables were labeled as "unclassified". For those files, we re-ran the algorithm with lower and lower thresholds (default was 0.3, decrease steps were of 0.05 and the minimum value was 0.15). We selected the first threshold with an "unclassified" rate lower than 5%. If there was no such threshold, we selected the threshold with the lowest "unclassified" rate. Nevertheless, manual examination of the remaining "unclassified" syllables showed still a considerable amount of real USVs. The shorter "unclassified" syllables were "simple" and the longer ones tended to be "multiple". As a result, and in order to assign each one of the syllables with one of the four basic labels (simple, down, up, multiple), we gave the "unclassified" syllables one of two labels: "simple" or "multiple". We used the median duration of all syllables in the database (35.3 ms) and set the syllables with a shorter duration as "simple" and with a longer duration as "multiple". In total there were 39,992 "unclassified" syllables of which 29,246 were labeled as simple (19.2% of the total simple population) and 10,746 were labeled as multiple (18.3% of total multiple). Next, to support an eight-label model, we split each one of the four labels into two. This was done using the median duration of all syllables with that were assigned the same label (simple: 27.6 ms, down: 48.1 ms, up: 50.7 ms and multiple: 96.3 ms). For example, "down" syllables that were shorter than 48.1 ms were labeled as Down-short and "up" syllables longer than 50.7 ms were labeled as Up-long.

*VoICE*[21]: The VoICE algorithm is based on hierarchical clustering. Running the algorithm on all syllables in the database was not feasible because of computation constraints. Therefore, 4000 syllables from different files were selected and the algorithm was applied to them. VoICE includes a manual phase (originally used for comparison) that was skipped. The results of the automatic phase are centroids that were further used to label all 346,632 syllables. The labeling was done using the same similarity measure that was used in the other parts of the VoICE algorithm.

*MUPET*[22]: MUPET uses a gammatone filter as part of the preprocessing. For the adapted version, we used 16 filters. As in the MSA algorithm, MUPET also contains a parsing phase. We loaded our syllable times instead of the built-in ones to maintain consistency. As the case with VoICE, the MUPET algorithm was not able to run on the full database, therefore we applied it on 5000 syllables. Then, we used the resulting centroids and the MUPET distance measure to label the rest of the syllables.

**Sample algorithms**. The four sample algorithms used to demonstrate the quantification framework are all based on the iMSA algorithm. The first algorithm generates two labels: one label (N: no jump) that is the result of merging the two simple labels (Simple-short and Simple-long) and another label (J: jump) that is the result of merging all other six labels. Besides the original version of the MSA algorithm, the second four-label algorithm contains both Simple-short and Simple-long labels, another label that contains all four Down/Up-short/long labels and a final label containing both the multiple labels (short and long). The final labeling algorithm is composed of five labels which are: Simple-short, Simple-long, Down-merged, Up-merged, and Multiple-merged. This diversity allows examining our framework for classifications with different numbers of labels and different distributions of syllables.

**Modeling the labeled sequences as Markov chains**. We apply a given labeling algorithm on all the syllables in our USV database and divide the labeled syllables into sequences, based on their ISI (with 160 ms as a threshold). These labeled sequences are discrete in time and space, where each discrete time-point has one of $N_c$ labels. We model these discrete sequences of random variables $(X_1, X_2, ..., X_n)$ as a Markov chain with a specific order d, and assume that the Markov process is

stationary and irreducible[35]. The realizations (observed values) of the random variables are denoted as: $x_1, x_2, …, x_n$. There are different ways to represent such Markov chains, for example as states with transition probabilities, or with suffix trees. We find that for our purpose suffix trees are more advantageous because they naturally represent the probability for the next syllable given the realization of the suffix and is easy to extend for increasing depth of the chain[36]. We define our suffix tree as a full tree where each leaf is associated with one suffix. The suffix is composed of the edge labels on the path from the root to the leaf. The number of leaves in the suffix tree is $N_c^d$ where $N_c$ is the number of labels and d is the depth of the tree. Therefore, for a labeling algorithm with four labels (like in Fig. 3): SUDM, and a tree with depth 2, the leaves will represent the following 16 suffixes: SS, SU, SD, SM, US, UU, UD,…, MU, MD, MM. In the leaves of the suffix tree we store two counts that are computed by scanning all the sequences, one-by-one: (1) We count how many times the branch leading to the leaf is visited. For example, for the leaf in Fig. 3 (e.g., syllable M in the branch DS, we keep N(DS) which counts the number of instances that the pair SD has appeared in all the sequences. (2) We also count N(M|DS) which is given by he number of instances that M appeared following the pair DS. Therefore $p(X_3 = M|X_2 = D, X_1 = S) = \frac{N(M|DS)}{N(DS)}$ and $p(X_2 = D, X_1 = S) = \frac{N(DS)}{\text{Total number of pairs}}$. It is important to note that every Markov chain of m-order could be represented as a 1st-order Markov chain (at the expense of increasing the state space). In our case the path of a branch on the tree from the source to the leaf is such a state, and the probability for each syllable at the leaf is the transition probability to a new state.

**Entropy rate calculation.** Our representation of a Markov chain as a suffix tree supports the calculation of the entropy rate based on the following equation:

Theorem 4.2.4 (Cover and Thomas, 2005) Let $\{X_i\}$ be a stationary Markov chain with stationary distribution $\mu$ and a transition matrix P. Let $X_1 \sim \mu$. Then the entropy rate is

$$H(Y) = -\sum_{ij} \mu_i P_{ij} \log P_{ij},$$

where $\mu_i$ are given by the probabilities of visiting the leaves and $P_{ij}$ are the transitions probabilities stored in the leaves (e.g., if we are in leaf DS the probability $p(M|DS)$ is the probability to move from state DS to state MD in the above example).

**SIS calculation.** We define the SIS as the MI (denoted as $I(X;Y)$) between the two random variables X and Y. The random variable X marks the next syllable $X_n$ and the random variable Y marks its prefix, which in our case for a given tree with depth D is of length D: $Y = X_{n-d}, …, X_{n-1}$. For the calculation of the MI, we use the following probability mass functions:

(1) $p(y) = p(X_{n-1}, …, X_{n-D})$, which is stored in the leaves as the probability to obtain each prefix.

(2) $p(x,y) = p(X_n, …, X_{n-D})$, which is calculated based on the law of total probability by multiplying each set of conditional probabilities $p(X_n|X_{n-1}, …, X_{n-D})$ with their corresponding $p(X_{n-1},…,X_{n-D})$.

(3) $p(x) = p(X_n)$, which is equivalent to the 0th-order probability mass function.

With these three probability mass functions, the MI $I(X;Y)$ is calculated as[35]

$$I(X;Y) = \sum_{x \in X} \sum_{y \in Y} p(x,y) \log \frac{p(x,y)}{p(x)p(y)},$$

In our case, this calculation for a tree with depth D is done by iterating over all possible $(D+1)$-tuples, such that for a $(D+1)$-tuple: $x_{n-d}, …, x_n$ (a sequence of observed values) the random variable Y is equal to $Y = x_{n-1}, …, x_{n-D}$ and the random variable X is equal to $X = x_n$. By performing the summation of the values for all $(D+1)$-tuples we compute the SIS score of the suffix tree.

In addition, instead of performing the summation, we can visualize each of these values separately and plot them, as can be seen in Figs. 4c, 7c, and Supplementary Fig. 5.

Finally, note that $I(X;Y)$ can also be viewed in two additional ways:

$$(1) \quad I(X;Y) = H(X) - H(X|Y),$$

where $H(X) = H(X_n)$ and $H(X|Y) = H(X_n|X_{n-1},…,X_{n-D})$

$$(2) \quad I(X;Y) = D_{KL}(p(x,y)||p(x)p(y)),$$

where $p(x, y)$, $p(x)$ and $p(y)$ are the same as previously defined and the $D_{KL}$ is defined by:

Definition (Cover and Thomas, 2005) The relative entropy or Kullback–Leibler distance[48] between two probability mass functions $p(x)$ and $q(x)$ is defined as:

$$D_{KL}(p||q) = \sum_{x \in X} p(x) \log \frac{p(x)}{q(x)},$$

Therefore, if X and Y are statistically independent then the MI $I(X; Y)$ is 0.

**Standard deviation of entropy rate and SIS calculations.** The entropy rates and SIS values are computed over 25 repetitions. In each repetition, 60% of the

sequences in the database are used to construct the suffix tree which the values are calculated from. The mean value of the 25 repetitions is plotted as a dot and the error bars mark 2 standard deviations.

**Validating the estimations with synthetic Markov models.** The ability to create a stochastic model that successfully captures the statistics of the labeled USVs depends on the amount of data that is available. As the order of the Markov model (and the depth of the equivalent suffix tree, D) grows, a larger amount of sequences is required for obtaining enough samples of each $(D+1)$-tuple. A case where during the process of constructing the suffix tree, there are several branches that are not reached due to a limited amount of data could lead to a wrong estimation of both the entropy rate and the SIS of the model. Therefore, to validate our estimation procedure, we have carried out a series of simulations in which we have constructed synthetic Markov models of various orders. For these synthetic Markov models, the analytical calculations of the entropy rate and SIS are obtained from the transition probability matrix. Given a Markov model represented by a suffix tree $T_{synt}$ with depth D and a set of conditional probabilities $p(x_n|x_{n-1},…, x_{n-D})$, the entropy rate is calculated as:

$$H(Y) = -\sum_{ij} \mu_i P_{ij} \log P_{ij},$$

The values of $P_{ij}$ are given by the conditional probabilities $p(x_n|x_{n-1}, … x_{n-D})$. Since the stationary distribution $\mu$ is defined by $\mu P = \mu$[49] then the values of the stationary distribution $\mu_i$ can be calculated as:

$$\mu = \frac{e}{\sum_i e_i},$$

where e is the eigenvector of the transition matrix P with an eigenvalue of 1.
Similarly, the SIS of the model can be calculated as

$$I(X;Y) = \sum_{x \in X} \sum_{y \in Y} p(x,y) \log \frac{p(x,y)}{p(x)p(y)},$$

where the three probability mass functions in this equation are given by: (1) $p(y)$, which is equivalent to $\mu_i$. (2) $p(x, y)$, which is calculated by multiplying each conditional probability $p(x_n|x_{n-1},…,x_{n-D})$ with its suffix probability $p(x_{n-1},…x_{n-D})$ (based on the law of total probability) and (3) $p(x)$, which can be obtained by performing D iterations $d = (D,.., 1)$ of $\mu = \frac{e}{\sum_i e_i}$, such that in each iteration e is the eigenvector with an eigenvalue of 1 of the transition matrix P corresponding to the stationary distribution of depth d.

After calculating the analytical values, the next step includes generating realizations of label sequences from these synthetic models. The realizations are generated by adding syllables one-by-one based the conditional probabilities of each label given the current suffix: $p(x_n|x_{n-1},…,x_{n-d})$. Next, a new suffix tree $T_{est}$ is created from the generated sequences and the entropy rate and SIS of $T_{est}$ are calculated. Finally, the values calculated from $T_{est}$ are compared to analytical ones and the difference between the estimated and analytical values embodies the estimation error.

**SIM algorithm.** For a given USV database, the SIM divides the sequences into two groups: training and test sets, with the same number of sequences in both sets. Every syllable in the database goes through the MUPET preprocessing procedure with 16 filters. This converts all the syllables into vectors with a length of 2016. The initial SIS is calculated for the training set, as well as the centroid of each cluster. The centroid is calculated as the mean of all syllables in the cluster. Then, the iterative process starts. In each iteration, a random vector V is created by generating values from a uniform distribution ranging between 0.9 and 1.1. This vector is used to perturb each centroid, one at a time. The perturbation is done as a dot product between the vector representing the centroid and V. For each perturbation, all syllables in the training set are reclustered and the SIS is calculated for the new labeling. After perturbing all centroids, there is an SIS value corresponding to each perturbation. The maximum value is selected and compared to the value before perturbation. If it is higher, the perturbation that achieved that value is applied and stored and a new random vector V is generated. If the maximum value is lower, then a "failure counter" is increased and no perturbation is done. Once the "failure counter" reaches the value of 5, the perturbation with the maximum SIS is applied, no matter if it is larger or smaller than the preperturbation SIS. Anytime a perturbation is applied, the "failure counter" is reset to 0.

This iterative process is repeated until the SIS of the training set converges. Once convergence is achieved, the SIM "replays" the same perturbation chain on the test set and the SIS in each step is calculated. The progress of the SIS on the training set and on the test set is then plotted.

**Statistics and reproducibility.** Parsing 385 recording sessions resulted with 346,632 syllables grouped into 33,481 sequences. The entropy rates and SIS values were computed over 25 repetitions. In each repetition, 60% of the sequences in the database are used to construct the suffix tree which the values are calculated from.

The mean value of the 25 repetitions is plotted as a dot and the error bars mark 2 standard deviations.

**Reporting summary**. Further information on research design is available in the Nature Research Reporting Summary linked to this article.

## Data availability

All data used to produce the figures and charts in this paper are available in Supplementary Data 1. The USV audio recordings ('wav' files) that were used in this study are available in mouseTube[47] with the group label "London Lab". Supplementary Table 1 lists these files and complementary meta data.

## Code availability

The computer code used in this study is available at: https://github.com/london-lab/MouseUSVs.

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

## Author Contributions

S.H. and M.L. conceived the study. B.W. and N.P. recorded the data used in the study. S.H. wrote the code, analyzed the data, and ran the simulations. S.H. and M.L. wrote the paper with input from all authors. M.L. supervised the study. All authors approved the paper prior to submission.

## Competing Interests

The authors declare no competing interests.

## Additional information

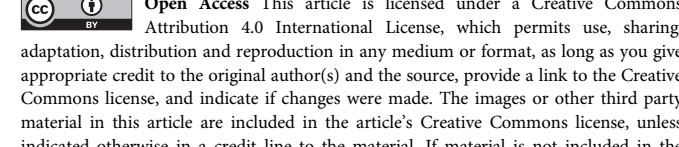

