## [Peer Review File · Communications Biology]

Reviewers' comments:

Reviewer #1 (Remarks to the Author):

The authors refer to a ground truth in the analysis of human speech, I am unfamiliar with what this represents, unless the authors are referring to metrics like those derived from EGG for vocal fold function – speech on the other hand is not definitively defined.

There is also some literature proposing that expert rater outcomes, like those derived by Scattoni et al., represent a 'ground truth' on which to compare, as it is what is currently used in the field to describe USVs.

However, the arguments put forth are reasonable and the reviewer agrees.

It is good the python toolkit is available online.

What were the strains etc of the mice included in the study.

Comparison of these three algorithmic approaches is a nice model for the study.

Really nice outcomes from this multi-algorithm approach. The suggestion that USV are not elicited in a temporally independent manner is of interest to the field and a worthy finding from this work.

There are clearly a number of steps to follow this work and I look forward to reading them in the future.

Reviewer #2 (Remarks to the Author):

Mouse ultrasonic vocalizations (USVs) are composed of complex (non-stereotyped, yet repetitive) syllable shapes and are emitted in patterned sequences (syntax). In order to decipher their social-behavioral significance, the field has developed a number of classification approaches, which employ different methods to categorize syllable shapes and therefore yield different results in terms of the syllable types and syntax that best describe a given set of recordings. In the absence of "ground truth" for syllable types (classes), it has been challenging for the field to move forward in deciphering the social communicative significance of different syllable shapes and syntax.

Here, the authors approach this problem by studying sequences of USVs in time using three different algorithms proposed in the literature to label each USV: MSE, VoICE, and MUPET. They use the labels for USVs produced by these three algorithms to study the structure in time and the paper makes the interesting point of proposing to measure the quality of classification of USVs as part of a sequence prediction problem: If we are able to learn the sequences correctly, and we are able to predict well what's the next USV class, then they assume they have better classified the USVs. Despite interest in this hypothesis and appreciation for the significant thought and effort that went into the analyses, I am concerned that the way in which the authors are computing the entropy and Kullback-Leibler (KL) divergence is not valid on Markov chains. This is described in Major Concerns 1-3 below. In addition, assumptions about the number of classes (8) and the expected syntax structure (pairs of syllables), also raise concerns about the validity of the conclusions. Examples of the strengths and weaknesses are described below.

Key Strengths.

1. Important topic for the field of social communication in general, and rodent USV communication in particular, where there is a pressing need for new approaches and the application of Information Theory to understand the underlying biology.
2. The paper is well written and provides a thorough description of the rationale, methodology and results. Terms are well-defined and the paper gives clear examples to illustrate aspects of the data analysis and results that otherwise might be missed by the reader.
3. They appropriately emphasize that this approach is an initial attempt at developing a new algorithm and has not yet been optimized in terms of computational complexity and processing requirements.
4. The conclusions are based on an impressive number of vocalizations (460K USVs, grouped into 44K sequences). As noted in the manuscript, this large number of vocalizations and sequences is necessary to be able to extract meaningful statistics on underlying structure. They deposit their vocalization files on MouseTube, making them accessible to the field, which is also a strength.
5. The authors use three popular algorithms to model the structure of the sequences and this is a benefit to the field because it provides an assessment of the outcomes of popular approaches (MSA, VoICE, MUPET) and a starting point for efficient collaboration across labs in the future.
6. Assessment of male-female courtship vocalizations is a strong and well-justified approach as males sing long and complex songs in the presence of a female and females rarely vocalize. It was not noted whether there was any evidence of female vocalizations (perhaps USVs occurring at the same time in different frequency bandwidths, suggesting dual vocalizers?). While one might recommend using female urine as a stimulus to avoid this potential confound, it is the experience of many labs that males vocalize much less frequently and consistently to female urine than an awake female. Thus, despite the potential for female vocalizations, I see this design as a justifiable choice for the types of analyses done here.

Major concerns.

1. Line 260: The KL divergence between two Markov models cannot simply be computed as suggested in this paper. In Markov chains, the outcome in time t depends on the previous outcomes. Because of this time-dependency, random processes are completely defined by the probability distributions at all times (think of $p(x_1, \dots, x_n)$). For a Markov chain of order 1, this is greatly simplified because $p(x_1, \dots, x_n) = p(x_1) * p(x_2 | x_1) * p(x_3 | x_2) * \dots * p(x_n | x_{n-1})$, since $p(x_n | x_{n-1}, \dots, x_1) = p(x_n | x_{n-1})$. Therefore, if you make assumptions, you can define the random process more easily in some cases. Here, the joint distribution $p(x_1, \dots, x_n)$ should be used to compute the entropy and KL divergence. This is usually difficult, and this is where there are concerns about the methods used in the paper:

A. The authors do not state their assumptions with respect to the Markov chains that they estimate. The only exception is the order.

B. They do not declare what distribution of the Markov chain they're using. I think they assume stationarity of the process (which is a very strong assumption, but it is OK if they state it because the problem is very hard otherwise) and that the Markov chain is irreducible, and then using the stationary distribution.

C. If, however, they are using $p(x_n | p_{x-1})$, then the formulas they are using do not hold.

2. Line 138: The authors only use 8 classes (labels) for possible USVs. This makes sense to estimate the sequence structure, as it will be easier to estimate the transitions between fewer classes (which is $O(n^2)$ with n being the number of classes). However, in the MUPET paper, it is established that under the MUPET algorithm, usually more classes are required. There is definitely a trade-off between the ease of sequence estimation and an accurate representation of USVs. My main concern here is that such a low number of classes might not represent well all the USVs (having many different USVs with the same label). In this case, the sequencing analysis loses validity.

3. Line 193: I have my doubts on using entropy to measure the predictive power. I would have done

this on the predicted labels of each algorithm instead. Entropy (as the text explains) is a measure of uncertainty (in this case, of the probability of having class label L at this time step given the m previous labels). If the structure of the sequences is low, you would have a higher entropy, but then you don't know if this is due to the USVs having no structure in time or because your classification method is performing poorly. Some of the problems are discussed in paragraphs starting in lines 233 and 242, but the problem identified here is not discussed.

4. More discussion is needed of prior work on mouse USV syntax structure in the field, as well as the data presented here, in terms of assumptions that are made about how often syllable motifs of size 2, 3, 4, repeating units (identical or not) are observed in mouse USVs. Here, the analysis looked at pairs of syllables, but there is evidence in the field for motifs of 3 or more syllables, which may not repeat the same syllable type (see Holy and Guo, 2005, Figure 6, Figure 7). What assumptions about syntax complexity (number of repeating syllables [i.e., AAA] and complexity of the repeating motif [i.e., ABC, ABC]) are being made in this analysis and how do they impact performance? Does the entropy analysis provide information on how many syllables are typically repeated in a sequences, how many types of motifs of various lengths are observed in this data set?

Minor comments

1. Line 129-130: iVOICE and iMUPET apply unsupervised methods to learn the "key" features. This is not true for MUPET: the filter bank is the feature extractor (which was designed), and the output features are used for clustering.

2. Line 138: It would be helpful to see a visual representation of the 8 syllable classes for each algorithm. This is more complicated for MSA because many different syllable shapes are contained under 'frequency step', but a representative example would be appropriate. Here the goal is for the reader to be able to understand the uniformity or diversity of syllable shapes that make up the different classes for each algorithm.

3. Did the males have prior sexual experience, which might enhance their vocalization rate?

4. How many times was each male recorded from? Were estrous females used?

5. Could be helpful to indicate with text why iVoICE is not compared along with iMSA and iMUPET throughout.

6. Given the better performance of iMSA, it could be helpful to indicate in the text why iMUPET was selected for optimization. Having worked with all 3 systems, I imagine this is a function of the ease of use, programming, syllable visualization, etc, but for non-experienced users to follow the logic (and to avoid incorrect assumptions for experienced users), making the rationale explicit would strengthen the paper. (e.g., Pg 17, line 332).

7. Would be helpful to include more references to prior analyses of syntax in mouse USVs.

Reviewer #4 (Remarks to the Author):

MAJOR POINTS

(1) This paper addresses the analysis of mouse courtship ultrasonic vocalizations (USVs). These can be divided into syllables which have been the subject of many attempts to classify them based on acoustic properties, leading to a welter of different schemes. The authors are interested in the sequences of syllables, in particular whether these are predictable, in the sense of birdsong or some aspects of human vocalizations. The argue that different classification schemes can be evaluated by the degree of predictability in syllable sequences produced by the classifications. The analysis is based on information theory and a very large dataset of the authors own recordings and those obtained from a collaborative web repository of mouse vocalizations. The analysis shows the existence of some sequential structure in the vocalizations analyzed, based on serial predictability of syllables, and also shows that the degree of predictability varies with the classification scheme (based on comparing three classification schemes from the literature). The results are interesting and make the authors'

point.

(2) The paper is based on the assumption that there should be a lawful sequence of syllable types, which their results support, and that this lawfulness reflects from production of different classes of syllables, as opposed to some incidental property of vocalizations (the motor program, adaptation or fatigue of the mechanism, etc.)

(3) The biological context of the work is not well developed. Vocalizations change with the mating activity of the mice involved (e.g. Matsumoto and Okanoya (2016)), so structure in the vocal stream could relate to the mating context (approach, sniffing, mounting, intromission, etc.) which is uncontrolled here. Or perhaps the comment should be "what was going on between the mice when the recordings used were recorded?" Are recordings from all stages of interaction included? Does predictability vary with the stage of interaction? Would the result differ if vocalizations from different stages were analyzed separately or if another variable measuring the stage of interaction were included in the analysis. These questions occur to me because the vocalizations used seem to be only swept tones, not harmonic stacks or noisy vocalizations reported by some authors. The degree of predictability may be influenced by a change in the pattern or characteristics of vocalization correlated with mating context. This is probably a source of predictability, especially for variables like duration and muddies the whole classification effort.

The previous paragraph does not mean that I don't like the analysis or disbelieve the results, I am just asking for the analysis to be put in context.

(4) The entropy rate analysis is fine and makes a point, for a reader with knowledge of information theory. It is the drop in entropy as more predictors are added which is important. This fact is a little obscure in the current presentation. Why not use mutual information where you can talk directly about information provided. by preceding context? It's also easier to describe the MI calculation and it is intuitive. The current description of the analysis strays into Markov models, which is fine theoretically, but it introduces a complex jargon (leafs, suffix trees, etc.) which is unnecessary to the exposition, and makes the paper hard to read. MI can be explained fairly simply, only requiring the use of joint and conditional probabilities (concepts that are already used). Also it would be useful in this section to have some estimate of the standard deviation of the entropy estimates, say from bootstrap. I suspect they are pretty small, given the dataset, but a single comment to that effect in the text would be helpful.

(5) lines 235-240, 245-250, 255-260 - if you use mutual information, this problem discussed in the first line range will be minimized, also true for the problem discussed in lines 245-250. The change in the upper limit of entropy will not (necessarily) change the MI. Of course, MI is closely related to entropy rate and also to the KL divergence method you consider in lines 255-260. I think they are they same, in fact, isn't this true?

MINOR POINTS

(6) Fig. 1F is not mentioned in the text. Is it important? Why is a cumulative distribution shown as opposed to a distribution, as for the others?

(7) Fig. 2 and others - text and figures too small (text should be same size as the text text). Lo-res reproduction does not allow blowing them up. Annoying.

line 202 - Figure 3C ??

(8) p. 11 - "reduction in entropy rate saturates at order-4 Markov models". This is not shown in the figure. I believe it, but just saying it in the text is mystifying. Perhaps add ('not shown')?

(9) lines 270-280 - why invent a new name for MI?

(10) line 330 - "analogous" spelled incorrectly

(11) line 365-66 - “. . . and also compared to the iMSA.” I don’t see this, the two seem to be about the same. Not sure why it’s important anyway.

(12) line 386-394 - This has been said, . . . and said, . . . and said.

Reviewer #1 (Remarks to the Author):

The authors refer to a ground truth in the analysis of human speech, I am unfamiliar with what this represents, unless the authors are referring to metrics like those derived from EGG for vocal fold function – speech on the other hand is not definitively defined.

We thank the reviewer for the comment. We only meant to indicate that in human speech it is possible (most of the time) to classify syllables into distinct known classes while in other species this may not be the case. We have modified the text to better indicate that (lines 53-55) and we dropped the term “ground-truth”.

There is also some literature proposing that expert rater outcomes, like those derived by Scattoni et al., represent a ‘ground truth’ on which to compare, as it is what is currently used in the field to describe USVs. However, the arguments put forth are reasonable and the reviewer agrees.

The paper by Scattoni et. al does indeed suggest an interesting way to classify syllables into classes. It is a good example of a classification that is based on predefined features rather than on the outcome of a clustering algorithm. In our work, the MSA algorithm is based on the same concept of predefined features. However, in the absence of any further knowledge (for example perceptual classes of the mice themselves), it is not obvious why one set of predefined features (i.e. Scattoni et al.) would be superior to another one (Chabout et. al. 2015) and would deserve to qualify as ‘ground truth’. In our manuscript we address this conflict by suggesting a different way to approach this problem.

It is good the python toolkit is available online.

We agree with the reviewer about this comment and hope that our toolkit will be beneficial for the scientific community.

What were the strains etc of the mice included in the study.

The description of the mice included in the study is now available in supplementary table 1. In addition, in the Material and Methods chapter it is stated that: “For the recordings performed in our lab, we used C57BL/6 male and female mice (8–12 weeks old).” (lines 561-562).

Comparison of these three algorithmic approaches is a nice model for the study. Really nice outcomes from this multi-algorithm approach. The suggestion that USV are not elicited in a temporally independent manner is of interest to the field and a worthy finding from this work. There are clearly a number of steps to follow this work and I look forward to reading them in the future.

We are grateful for these kind words and are highly excited to carry on this work too.

Reviewer #2 (Remarks to the Author):

Mouse ultrasonic vocalizations (USVs) are composed of complex (non-stereotyped, yet repetitive) syllable shapes and are emitted in patterned sequences (syntax). In order to decipher their social-behavioral significance, the field has developed a number of classification approaches, which employ different methods to categorize syllable shapes and therefore yield different results in terms of the syllable types and syntax that best describe a given set of recordings. In the absence of “ground truth” for syllable types (classes), it has been challenging for the field to move forward in deciphering the social communicative significance of different syllable shapes and syntax.

Here, the authors approach this problem by studying sequences of USVs in time using three different algorithms proposed in the literature to label each USV: MSE, VoICE, and MUPET. They use the labels for USVs produced by these three algorithms to study the structure in time and the paper makes the interesting point of proposing to measure the quality of classification of USVs as part of a sequence prediction problem: If we are able to learn the sequences correctly, and we are able to predict well what's the next USV class, then they assume they have better classified the USVs. Despite interest in this hypothesis and appreciation for the significant thought and effort that went into the analyses, I am concerned that the way in which the authors are computing the entropy and Kullback-Leibler (KL) divergence is not valid on Markov chains. This is described in Major Concerns 1-3 below. In addition, assumptions about the number of classes (8) and the expected syntax structure (pairs of syllables), also raise concerns about the validity of the conclusions. Examples of the strengths and weaknesses are described below.

We thank the reviewer for the interest and appreciate the comprehensive review. We provide detailed responses for the stated concerns beneath each corresponding point. We addressed the concern regarding entropy and SIS computation (which in the revised manuscript is computed based on mutual information instead of KL-divergence) by clarifying our assumptions and calculations. Furthermore, we added a new figure that is focused on the number of classes and its effect on our findings.

Key Strengths.

1. Important topic for the field of social communication in general, and rodent USV communication in particular, where there is a pressing need for new approaches and the application of Information Theory to understand the underlying biology.
2. The paper is well written and provides a thorough description of the rationale, methodology and results. Terms are well-defined and the paper gives clear examples to illustrate aspects of the data analysis and results that otherwise might be missed by the reader.
3. They appropriately emphasize that this approach is an initial attempt at developing a new algorithm and has not yet been optimized in terms of computational complexity and processing requirements.
4. The conclusions are based on an impressive number of vocalizations (460K USVs, grouped into 44K sequences). As noted in the manuscript, this large number of vocalizations and

sequences is necessary to be able to extract meaningful statistics on underlying structure. They deposit their vocalization files on MouseTube, making them accessible to the field, which is also a strength.

5. The authors use three popular algorithms to model the structure of the sequences and this is a benefit to the field because it provides an assessment of the outcomes of popular approaches (MSA, VoICE, MUPET) and a starting point for efficient collaboration across labs in the future.

6. Assessment of male-female courtship vocalizations is a strong and well-justified approach as males sing long and complex songs in the presence of a female and females rarely vocalize. It was not noted whether there was any evidence of female vocalizations (perhaps USVs occurring at the same time in different frequency bandwidths, suggesting dual vocalizers?). While one might recommend using female urine as a stimulus to avoid this potential confound, it is the experience of many labs that males vocalize much less frequently and consistently to female urine than an awake female. Thus, despite the potential for female vocalizations, I see this design as a justifiable choice for the types of analyses done here.

We thank the reviewer for the supporting comments. Indeed, we have chosen the male-female setup in order to maximize the number of USVs recorded. We have mentioned the possibility of a low occurrence of female vocalizations as a minor confound in the discussion (lines 482-486).

Major concerns.

1. Line 260: The KL divergence between two Markov models cannot simply be computed as suggested in this paper. In Markov chains, the outcome in time t depends on the previous outcomes. Because of this time-dependency, random processes are completely defined by the probability distributions at all times (think of $p(x_1, \dots, x_n)$). For a Markov chain of 1st-order, this is greatly simplified because $p(x_1, \dots, x_n) = p(x_1) * p(x_2 | x_1) * p(x_3 | x_2) * \dots * p(x_n | x_{n-1})$, since $p(x_n | x_{n-1}, \dots, x_1) = p(x_n | x_{n-1})$. Therefore, if you make assumptions, you can define the random process more easily in some cases. Here, the joint distribution $p(x_1, \dots, x_n)$ should be used to compute the entropy and KL divergence. This is usually difficult, and this is where there are concerns about the methods used in the paper:

A. The authors do not state their assumptions with respect to the Markov chains that they estimate. The only exception is the order.

B. They do not declare what distribution of the Markov chain they're using. I think they assume stationarity of the process (which is a very strong assumption, but it is OK if they state it because the problem is very hard otherwise) and that the Markov chain is irreducible, and then using the stationary distribution.

C. If, however, they are using $p(x_n | p_{x-1})$, then the formulas they are using do not hold.

We thank the reviewer for this profound comment.

With respect to the assumptions we make, indeed we assume stationarity and irreducibility (we agree that these are strict assumptions, however, as agreed by the reviewer, these are reasonable assumptions to make in order to facilitate this analysis in the context of complex behavioral scenarios we consider here). These assumptions are now made clear in the text (please see lines 200-210).

With respect to comment C, we have not used the conditional probabilities in the equation that was used to calculate the entropy, which is indeed a wrong thing to do. We have now stated our entropy calculations much more clearly (lines 210-221 and 665-700; and also please see the explanation below).

In addition, to further validate our estimation procedure, in this revision we have carried out a series of simulations of synthetic Markov models of various orders. For each, we calculated the entropy rate analytically from the transition probability matrix (in the same way Cover and Thomas demonstrate in page 73 (example 4.1.1) for the two-state Markov chain example; Note that the stationary probabilities of visiting each state can also be computed from the transition matrix). We then generated realizations of label sequences from these synthetic models and ran our estimation algorithm on them. A comparison between the results of the estimation process and of the analytically computed entropy rate is now shown in figure S4. This figure demonstrates that our algorithm converges to the analytical solutions in all cases tested.

With regard to our entropy and SIS estimation procedure, we consider only processes of discrete-time (individual syllables) and state space. There are different ways to represent such Markov chains, for example, as states with transition probabilities, or with suffix trees. We find that, for our purpose, suffix trees are more advantageous. Suffix trees naturally represent the probability for the next syllable given the realization of the suffix, and they are also easy to extend for increasing depth of the chain (for discussion of suffix trees implementation see for example - Gabadinho, Alexis, and Gilbert Ritschard *Journal of statistical software* 72, no. 3 (2016): 1-39). In the leaves of the suffix tree, we store two counts: (1) We count how many times the branch leading to the leaf is visited. For example, for the leaf in figure 3 (e.g. in the branch DS), we keep $N(DS)$ which counts the number of instances that the pair SD has appeared in all the sequences (note that in our notation a suffix is read backwards in time, from right to left, which supports the easy extension of the tree for larger depths). Additionally, we count $N(M|DS)$ which is given by the number of instances that M appeared following the pair SD. Therefore $p(x_3 = M|x_2 = D, x_1 = S) = \frac{N(M|DS)}{N(DS)}$ and $p(x_2 = D, x_1 = S) = \frac{N(DS)}{\text{Total number of pairs}}$. It is important to note that every Markov chain of m-order could be represented as a 1st-order Markov chain (at the expense of increasing the state space). In our case, the path of a branch on the tree from the source to the leaf is such a state, and the probability for each syllable at the leaf is the transition probability to a new state.

For calculating the entropy rate, therefore, we can use equation 4.27 from page 77 of Cover and Thomas (2005):

$$H(Y) = - \sum_{ij} \mu_i P_{ij} \log P_{ij}$$

Where μ_i are given by the probabilities of visiting the leaves, and P_{ij} are the transition probabilities stored in the leaves (e.g., if we are in leaf DS, then $p(M|DS)$ is the probability to move from state DS to state MD).

In terms of the SIS, in the revised manuscript we define the SIS as the mutual information (denoted as $I(X; Y)$) between the two random variables X and Y. The random variable X marks the next syllable X_n and the random variable Y marks its prefix, which in our case for a given tree with depth D is of length D: X_{n-d}, \dots, X_{n-1} . For the calculation of the mutual information, we use the following probability mass functions:

- (1) $p(y) = p(X_{n-1}, \dots, X_{n-D})$, which is stored in the leaves as the probability to obtain each suffix.
- (2) $p(x, y) = p(X_n, \dots, X_{n-D})$, which is calculated by multiplying each set of conditional probabilities $p(X_n | X_{n-1}, \dots, X_{n-D})$ with their corresponding $p(X_{n-1}, \dots, X_{n-D})$.
- (3) $p(x) = p(X_n)$, which is equivalent to the 0th-order probability mass function.

With these three probability mass functions, the mutual information $I(X; Y)$ is calculated as follows (page 20 in Cover and Thomas, 2005):

$$I(X; Y) = \sum_{x \in X} \sum_{y \in Y} p(x, y) \log \frac{p(x, y)}{p(x)p(y)} \quad (\text{equation 2.28})$$

2. Line 138: The authors only use 8 classes (labels) for possible USVs. This makes sense to estimate the sequence structure, as it will be easier to estimate the transitions between fewer classes (which is $O(n^2)$ with n being the number of classes). However, in the MUPET paper, it is established that under the MUPET algorithm, usually more classes are required. There is definitely a trade-off between the ease of sequence estimation and an accurate representation of USVs. My main concern here is that such a low number of classes might not represent well all the USVs (having many different USVs with the same label). In this case, the sequencing analysis loses validity.

The reviewer point is well taken. To address this point, we have performed additional analyses on our database (lines 358-367) using the iMUPET algorithm while varying the number of classes (N_c), as suggested, and present the findings in the new figure 5 and accompanying text in lines 367-380.

In short, we have calculated the SIS for the iMUPET algorithm for N_c ranging from 8 to 64 for a tree of depth 1; and N_c of 8 and 16 for a tree of depth 2 (figure 5 and see also figure S6). For larger N_c we could not obtain a valid entropy estimation (Strong et al., 1998; Paninski, 2003). We show that in these cases, the SIS does increase as a function of N_c , which means that increasing the number of classes does improve the predictability of the next syllable given the suffix. Notice that the increase of SIS with N_c comes at the price of an exponential increase in the complexity of the model. It is therefore interesting to consider how the increase in the SIS is compared to the number of bits required to encode each syllable. This is obtained by normalizing SIS by $\log_2(N_c)$ and, as shown in figure 5C, this normalized SIS does not have a clear dependency on the number of classes. This is now further discussed in the discussion section.

3. Line 193: I have my doubts on using entropy to measure the predictive power. I would have done this on the predicted labels of each algorithm instead. Entropy (as the text explains) is a measure of uncertainty (in this case, of the probability of having class label L at this time step

given the m previous labels). If the structure of the sequences is low, you would have higher entropy, but then you don't know if this is due to the USVs having no structure in time or because your classification method is performing poorly. Some of the problems are discussed in paragraphs starting in lines 233 and 242, but the problem identified here is not discussed.

We have added new lines (267-271 and 312-218) to account for the point raised towards the end of this comment (regarding structureless data vs. poorly performing algorithm).

Regarding the first part of the comment: firstly, we do all the entropy calculations on labels; secondly, the entropy as a measure for predictive power is only suggested as a starting point, and after highlighting the drawbacks for using it, we suggest utilizing the SIS (the mutual information between the suffix and the next label) for comparing between algorithms; lastly, we performed this analysis only after we had found that the USV sequences have an underlying temporal structure, even at basic features (such as the syllable duration; figure 1G), and that all the three algorithms examined are able to capture some of its structure (figure 2C). Following that, we claim that the ability to account for a high proportion of this structure is an indication for a good classification algorithm.

4. More discussion is needed of prior work on mouse USV syntax structure in the field, as well as the data presented here, in terms of assumptions that are made about how often syllable motifs of size 2, 3, 4, repeating units (identical or not) are observed in mouse USVs. Here, the analysis looked at pairs of syllables, but there is evidence in the field for motifs of 3 or more syllables, which may not repeat the same syllable type (see Holy and Guo, 2005, Figure 6, Figure 7). What assumptions about syntax complexity (number of repeating syllables [i.e., AAA] and complexity of the repeating motif [i.e., ABC, ABC]) are being made in this analysis and how do they impact performance? Does the entropy analysis provide information on how many syllables are typically repeated in a sequence, how many types of motifs of various lengths are observed in this data set?

We have expanded the text about motifs in the discussion section as requested (see lines 495-509). With regard to the questions raised, the methods described in the manuscript examine the difference between actual occurrences of motifs with a given length (determined by the depth of the suffix tree) and their expected occurrence under the assumption that the next syllable and the suffix are statistically independent. It does not carry any further assumptions regarding the complexity of the motif or the repetitions of syllables inside the motif. We claim that this is the right way to quantify motifs, because a purely high occurrence rate of a motif is not meaningful if it is completely expected by the probabilities of its individual components.

In this work we focused on motifs of length 2 and following the reviewer comment we also added motifs of length 3 (figure S5, panel B). While examining the motifs of length 2, we noticed that the most dominant ones included repeating syllables. This finding was a feature of both the iMSA and iMUPET algorithms.

In addition, we also emphasized that the SIS measure itself does not directly indicate what are the motifs that exist in the sequences. However, if the probability of a specific motif for appearing is higher than expected (assuming statistical independence of the suffix and the next syllable),

then this will contribute to an increase in the SIS. Therefore, if the SIS increases with the depth of the suffix tree, it indicates that motifs of that order appeared in the data. For finding those motifs, we looked at the deviation of the motif probability from the expected one (figure S5). This analysis highlights the motifs that most contribute to the increase in SIS.

Finally, we have now noted in the text that our method could be applied for an arbitrary motif length, but as the length increases, the amount of sequences required increases exponentially in order to collect enough statistics.

Minor comments

1. Line 129-130: iVOICE and iMUPET apply unsupervised methods to learn the "key" features. This is not true for MUPET: the filter bank is the feature extractor (which was designed), and the output features are used for clustering.

We thank the reviewer for this comment. We updated the text (line 137) and figure 2A based on this comment.

2. Line 138: It would be helpful to see a visual representation of the 8 syllable classes for each algorithm. This is more complicated for MSA because many different syllable shapes are contained under 'frequency step', but a representative example would be appropriate. Here the goal is for the reader to be able to understand the uniformity or diversity of syllable shapes that make up the different classes for each algorithm.

We agree with this comment. We added a new figure that provides a visualization of syllables from each one of the 8 classes (figure S2).

3. Did the males have prior sexual experience, which might enhance their vocalization rate?
4. How many times was each male recorded from? Were estrous females used?

We thank the reviewer for raising these two points. We have added a table and a supplementary excel file, listing all the mice participating in this study, their sexual experience and vocalization rate. In addition, we added the following description to the Materials and Methods part of our manuscript: "Our approach in this work was to include different males, with different sexual experience and to search for the underlying common sequence structure. Therefore, many of the males were recorded several times and the sessions were scheduled independently of the females' estrous."

5. Could be helpful to indicate with text why iVOICE is not compared along with iMSA and iMUPET throughout.

We limited the comparison at times to highlight specific points and avoid repetitions. However, we have addressed this comment by making sure that whenever a comparison is made, all algorithms are part of the comparison, either in the main body or in the supplementary information. Regarding figure 7A, the y-axis of both panels is focused on the SIS values

obtained by the SIM algorithm. Therefore, the SIS value of the iVoICE algorithm is not visible in these panels.

6. Given the better performance of iMSA, it could be helpful to indicate in the text why iMUPET was selected for optimization. Having worked with all 3 systems, I imagine this is a function of the ease of use, programming, syllable visualization, etc, but for non-experienced users to follow the logic (and to avoid incorrect assumptions for experienced users), making the rationale explicit would strengthen the paper. (e.g., Pg 17, line 332).

Although the points raised by the reviewer are well deserved, here we focus on whether SIS, as a constraint, might improve performance or degrade it. For that purpose, it is instructive to compare the algorithm that achieves the second-best SIS score to the two algorithms with the highest and lowest SIS, as this makes the scale of improvement more natural. We have clarified this better in the text (lines 407-410).

7. Would be helpful to include more references to prior analyses of syntax in mouse USVs. We thank the reviewer for the comment. We added references to the following papers: (Kalcounis-Rueppell et al., 2006), (Von Merten et al., 2014).

Reviewer #4 (Remarks to the Author):

MAJOR POINTS

(1) This paper addresses the analysis of mouse courtship ultrasonic vocalizations (USVs). These can be divided into syllables which have been the subject of many attempts to classify them based on acoustic properties, leading to a welter of different schemes. The authors are interested in the sequences of syllables, in particular whether these are predictable, in the sense of birdsong or some aspects of human vocalizations. They argue that different classification schemes can be evaluated by the degree of predictability in syllable sequences produced by the classifications. The analysis is based on information theory and a very large dataset of the authors own recordings and those obtained from a collaborative web repository of mouse vocalizations. The analysis shows the existence of some sequential structure in the vocalizations analyzed, based on serial predictability of syllables, and also shows that the degree of predictability varies with the classification scheme (based on comparing three classification schemes from the literature). The results are interesting and make the authors' point.

(2) The paper is based on the assumption that there should be a lawful sequence of syllable types, which their results support, and that this lawfulness reflects from production of different classes of syllables, as opposed to some incidental property of vocalizations (the motor program, adaptation or fatigue of the mechanism, etc.)

We thank the reviewer for this elegant summary and for the kind words.

(3) The biological context of the work is not well developed. Vocalizations change with the mating activity of the mice involved (e.g. Matsumoto and Okanoya (2016)), so structure in the vocal stream could relate to the mating context (approach, sniffing, mounting, intromission, etc.) which is uncontrolled here. Or perhaps the comment should be "what was going on between the mice when the recordings used were recorded?" Are recordings from all stages of interaction included? Does predictability vary with the stage of interaction? Would the result differ if vocalizations from different stages were analyzed separately or if another variable measuring the stage of interaction were included in the analysis. These questions occur to me because the vocalizations used seem to be only swept tones, not harmonic stacks or noisy vocalizations reported by some authors. The degree of predictability may be influenced by a change in the pattern or characteristics of vocalization correlated with mating context. This is probably a source of predictability, especially for variables like duration and muddies the whole classification effort.

The previous paragraph does not mean that I don't like the analysis or disbelieve the results, I am just asking for the analysis to be put in context.

We thank the reviewer for raising this interesting point. We agree with the reviewer that behavior during social interaction affects the composition of USVs and may also affect their temporal correlation within sequences. We have included a discussion of these points (lines: 479-493). In

short, our database is composed of USVs recorded in a male-female interaction setup during **all** phases of interaction. However, these USVs were **not** annotated with respect to what phase they were recorded in. Therefore, we have not addressed whether subdividing the data according to the interaction phase affects the predictability in the present study. We do agree that careful tracking of the behavioral context for each USV, can unravel the even finer structure of the USV syntax, and will make further progress towards the goal of understanding the information USVs carry about behavior.

(4) The entropy rate analysis is fine and makes a point, for a reader with knowledge of information theory. It is the drop in entropy as more predictors are added which is important. This fact is a little obscure in the current presentation. Why not use mutual information where you can talk directly about information provided. by preceding context? It's also easier to describe the MI calculation and it is intuitive. The current description of the analysis strays into Markov models, which is fine theoretically, but it introduces a complex jargon (leafs, suffix trees, etc.) which is unnecessary to the exposition, and makes the paper hard to read. MI can be explained fairly simply, only requiring the use of joint and conditional probabilities (concepts that are already used).

We thank the reviewer for this comment and accept it. Following the comment, we have modified the manuscript to use mutual information as the quantity to consider. We note that for the case of pairs of syllables, the mutual information is identical to KL divergence (Cover and Thomas, 2005). However, for more than two variables the generalization of the mutual information needs a careful definition. Let X_k be the random variable which takes one of the possible labels for the syllable at place k in the sequence, then for a sequence of labelled syllables (X_{n-D}, \dots, X_n) we use $I(X_n; X_{n-1}, \dots, X_{n-D})$ as the measure for the SIS, namely the mutual information between the next syllable and its prefix (see lines 288-297 for more details). Note that the mutual information of multiple variables such as $I(X; Y; Z)$ is not intuitive and does not capture exactly what we look for in the SIS. For example, $I(X; Y; Z)$ can be negative, while the SIS is guaranteed to be non-negative.

This change has, therefore, no effect on the estimation involving pairs of variables. The other estimations were recalculated and updated in the figures and text. The explanation above was added in the main text and the Methods section (lines: 288-297, 701-729).

Also, it would be useful in this section to have some estimate of the standard deviation of the entropy estimates, say from bootstrap. I suspect they are pretty small, given the dataset, but a single comment to that effect in the text would be helpful.

We performed the suggested estimation of the standard deviation of the entropy calculations and added it to figures 4, 5, 7, and S4.

(5) lines 235-240, 245-250, 255-260 - if you use mutual information, this problem discussed in the first line range will be minimized, also true for the problem discussed in lines 245-250. The change in the upper limit of entropy will not (necessarily) change the MI. Of course, MI is closely

related to entropy rate and also to the KL divergence method you consider in lines 255-260. I think they are the same, in fact, isn't this true?

We thank the reviewer for this important comment.

The final statement made by the reviewer is indeed correct, but only for the case of two variables. However, in our view, one has to have significant familiarity with entropy and MI in order to understand how it solves the issues raised in the manuscript (and mentioned by the reviewer). Risking being over didactic, we, therefore, prefer to take the reader step by step over the pitfalls and conclude with our final choice of using the MI as a measure for algorithm comparison. Please note that not all problems are solved by using MI. For example, the relationship between MI and the number of classes used is nonobvious, as explained in the response to comment 4 made by reviewer #2 and now covered in the revised manuscript with an additional figure (figure 5).

MINOR POINTS

(6) Fig. 1F is not mentioned in the text. Is it important? Why is a cumulative distribution shown as opposed to a distribution, as for the others?

We thank the reviewer for the comment. The panel is now mentioned in the manuscript (lines 113-116) and presents a distribution (not a cumulative one) like the other panels.

(7) Fig. 2 and others - text and figures too small (text should be same size as the text text). Low-res reproduction does not allow blowing them up. Annoying.

We apologize if the reviewer found it difficult to see the figure properly. We have made efforts to make them as clear as possible in the revised manuscript.

line 202 - Figure 3C ??

Correct. Changed to Figure 3C.

(8) p. 11 - "reduction in entropy rate saturates at order-4 Markov models". This is not shown in the figure. I believe it, but just saying it in the text is mystifying. Perhaps add ('not shown')?

We updated the text based on the suggestion.

(9) lines 270-280 - why invent a new name for MI?

We thank the reviewer for raising this point. We have now made sure that the text clearly states (lines 288-297) that the revised SIS refers to the specific measure of mutual information between the next syllable and its prefix. The definition of the SIS is useful to differentiate this measure from other possible measures that are based on MI, e.g., the MI between each one of the successive syllables in a sequence.

(10) line 330 - "analogous" spelled incorrectly

Corrected.

(11) line 365-66 - “. . . and also compared to the iMSA.” I don't see this, the two seem to be about the same. Not sure why it's important anyway.

We have corrected these lines.

(12) line 386-394 - This has been said, . . . and said, . . . and said.

We have removed these lines.

Reviewers' comments:

Reviewer #1 (Remarks to the Author):

I am satisfied with the response to reviewers

Reviewer #2 (Remarks to the Author):

Hertz and colleagues present an extensively revised manuscript that includes additional analyses and figures and additional descriptions of the results, methods, and interpretations of the findings. These additions strengthen the manuscript and satisfy all concerns related to mouse USVs and the neurobiological interpretations of the findings. Because my expertise is not in computer or information sciences, I am unable to adequately comment on the technical aspects of the algorithm design and implementation choices. I recommend a review of these methodological choices by a computer scientist or engineer with the appropriate expertise.

Reviewer #3 (Remarks to the Author):

The authors have responded well to the review comments. I have no further comments.

Reviewer #4 (Remarks to the Author):

General Comments

=====

- The paper is well written and it is easy to read.
- The experimental setup is clear, as well as the experiments.
- The comments in the previous round of reviews have been addressed. The math looks sounds, and the assumptions are clearly stated and the calculations are backed by plots and the tree structure they assume for the Markov model.
- The major concern regarding the use of only 8 classes has been justified with the follow-up using 32 classes from MUPET.

Major comments

=====

The authors mention throughout the paper that the different methods (MSA, VoiCE, MUPET) classify USVs into classes. This is not appropriate, as classification tasks (in statistics/machine learning) are supervised learning problems in which class labels exist. However, this is an unsupervised problem where class labels are not known and therefore, "classification" throughout the manuscript should be changed for "clustering", while "classes" should be changed for "clusters".

I would like to see further analysis on the output of the algorithms employed. There seems to be a disconnection between the sequence modeling and the output of each of these algorithms. For example, k-means used in MUPET produces clusters that have equivalent radii, which may introduce biases into the decision boundaries. Does this have an impact your method? See final remarks at the end of the review.

Minor comments

=====

- (Line 62) A new paper has been published on unsupervised learning of USVs. Maybe you can consider including it: <https://arxiv.org/pdf/2003.05897.pdf> Note that this paper deviates from VoiCE/MUPET since newer clustering techniques are used, showing advantages over k-means.
- (Line 107) When fitting a distribution over data, you should not use a R^2 statistic for the goodness-of-fit, but rather a test for fitting distributions. Wikipedia has a nice list here: https://en.wikipedia.org/wiki/Goodness_of_fit#Fit_of_distributions
- (Line 116) Same as above, use a different test.
- (Line 168) Remove the word "large" (what is a large deviation vs. a deviation?)
- (Line 289) What does a semi-colon ; mean in this notation?
- (Line 340) Could be useful mention that this is the definition of KL divergence (for completeness).
- (Line 349) The use of "expected" seems inappropriate: What (or why) is expected?
- (Line 387) The problem with the clustering algorithms not only lies on the lack of separability between clusters, but rather that this problem does not have a natural measure of similarity between USVs. This seems to be the major difficulty to find good representations through unsupervised learning.
- (Line 409) bonded -> bounded?
- (Line 429) extremely -> low. Extreme seems arbitrary.
- Figure 7: Both plots in A should have the same scale in the y-axis, so that they can be more easily compared.
- Paragraph starting at 445: The authors mention only backward-dependency in time. Seems reasonable that there could be backward and forward dependency.
- (Line 467) This problem does not necessarily have a "true" classification. Moreover, a wider set of features might not be needed, but a better measure of similarity between USVs to discriminate between them.
- (Line 472) True separation is not technically correct. USVs clusters are either separable or not.
- (Line 497) I do not agree with the the "unexpected" motifs comment. The occurrence probabilities can be very noisy due to biases introduced in the clustering algorithms, which is already known by the authors: the different algorithms (MSA, VoiCE, MUPET) do cluster different USVs very differently. Knowing this, what is expected? The expected probabilities depend on the bias introduced by each algorithm.
- (Line 512) I think that the word "demonstrated" is too strong in this setting, which is empirical. I would change it for "suggests".
- (Line 514) After "original one", a new paragraph should start. In that line, you start discussing the computational complexity of your implementation, which is different than the previous idea and might create confusion for readers that are less computationally-inclined.
- (Lines 543-548) The authors should also mention the trainability of increasing the number of classes. More data is needed to not overfit. Moreover, you use maximum likelihood (line 685) to estimate the transition probabilities. It would be a good addition to include how much data is needed.
- (Line 689) Either use X_i or x_i for random variables, but not both. Changing the notation might be confusing for some readers.
- (Definitions and theorems) Remove the equation numbers coming from the reference from which you obtained them, as this is not part of the format of your paper.
- (Line 703) Add "Y =" before $X_{\{n-d\}}$...
- (Line 705) which is the stored -> which is stored
- (Line 707) Add that you use total probability to compute the joint density.
- (Line 725) A note about how MI measures independence of random variables would be useful.
- (Line 751) Use a notation for vectors such as underbar or bold to make clear that you are dividing entry-wise, otherwise it is a bit hard to read.

Final remarks - A thought experiment

=====

Let us assume we have a Markovian sequence of order p . That sequence is corrupted by a structured noise (could be Markovian noise) which changes the labels of USVs with some probability. This could potentially create new motifs not originally existing in our sequence.

Let us assume that we have an estimate of the cluster to which each symbol belongs to (i.e. your initialization of the SIS) obtained from the noisy version of the sequence, and that you run SIS and find motifs.

What have you learned?

With this thought experiment, I am trying to point out that it is not straightforward to claim what has one "learned" (in the statistical learning sense) with such an approach. When the unsupervised learning algorithms give a biased sequence, one might learn the predictability of the next symbol of the "true signal" (if it exists) plus noise (introduced by the bias of the initial selection of clusters). A note on this topic in the discussion would be appreciated. Moreover, the authors could include that this is partly a limitation of using Markov Models (instead of, for example, Hidden Markov Models), since the observation is assumed to be the actual label.

Reviewer #4 (Remarks to the Author):

General Comments

=====

- The paper is well written and it is easy to read.
- The experimental setup is clear, as well as the experiments.
- The comments in the previous round of reviews have been addressed. The math looks sounds, and the assumptions are clearly stated and the calculations are backed by plots and the tree structure they assume for the Markov model.
- The major concern regarding the use of only 8 classes has been justified with the follow-up using 32 classes from MUPET.

Major comments

=====

The authors mention throughout the paper that the different methods (MSA, VoiCE, MUPET) classify USVs into classes. This is not appropriate, as classification tasks (in statistics/machine learning) are supervised learning problems in which class labels exist. However, this is an unsupervised problem where class labels are not known and therefore, "classification" throughout the manuscript should be changed for "clustering", while "classes" should be changed for "clusters".

We thank the reviewer for this comment. We have carefully changed classes into clusters where appropriate, but in some other places we kept the original notation and, in some places, we used the term labeling. We explain this below.

Firstly, we agree that strictly speaking the reviewer is correct and since this is an unsupervised process, clustering is more appropriate. However, we note that in all USV literature the process used for labeling USVs is called "classification algorithm" and we used it for consistency. Note also that for methods such as MSA (and several other manual approaches) where the possible labels are predetermined and strict rules are used to assign labels to USVs, "clustering" is less relevant. For example, in MSA the number of pitch-jumps and their direction exclusively determine the label. Since clustering does not fit all algorithms, we have now chosen the term labeling as the common action applied by the family of algorithms, and use clustering when addressing a specific algorithm that indeed clusters USVs, such as MUPET, VoiCE, and SIM.

Secondly, there is also the possibility that USVs belong to true classes. For example that both the neural mechanisms that are used to generate USVs and the ones used to perceive them have a naturally disjoint representation of classes, similar to what we believe we, as humans, have for the syllables that we use in speech. In that case, the term classification would be appropriate to describe labeling USVs by their true classes. When discussing this option in the text we, therefore, use the terms classes and classification.

I would like to see further analysis on the output of the algorithms employed. There seems to be a disconnection between the sequence modeling and the output of each of these algorithms. For example, k-means used in MUPET produces clusters that have equivalent radii, which may

introduce biases into the decision boundaries. Does this have an impact your method? See final remarks at the end of the review.

We thank the reviewer for this comment and added text in the discussion to address it (lines 510-519).

The disconnection between the sequence modeling and the output of the clustering algorithm is deliberate. As we note in the text, all known clustering algorithms applied to USVs consider each syllable at a time, independent of its position in a sequence, or the preceding syllables. Indeed this process can have biases and we highlight in figure 6a that the clustering may introduce ambiguity just as the reviewer suggests. Intuitively, because this bias of the algorithm is independent of the statistical structure of the sequence, it can be regarded as noise. As such, it should reduce the SIS because the dependence of the next syllable on its prefix is now “more random”.

Formally, one way to think about this bias is, through the setup of an information channel with a source given by a Markov chain with a finite alphabet. The Markov chain (representing the true underlying process generating the USVs) generates labels which are transmitted through the channel where the label can be corrupted by the noise of the channel (i.e. a mixing probability distribution where each label can be switched with some probability to another label) and the observer at the output of the channel receives these corrupted labels. Calculations of entropies in this setup, which is a specific case of HMMs (as suggested by the reviewer), happen to be notoriously difficult. Even the case of a first-order Markov chain over a binary alphabet with a binary symmetric channel is challenging (see for example Zuk et al. (2005) and Jacquet et al. (2008)). In fact, as far as we can find, all the studies addressing this problem have tried to compute or estimate the entropy of the output of the channel with respect to the noise and Markov model parameters. None have ever tried to compute the mutual information between the suffix and the next syllable as required for the SIS. We, therefore, think that formal analysis of this problem should be reserved for future efforts to develop the SIS approach.

Zuk O, Kanter I, Domany E. The entropy of a binary hidden Markov process. Journal of statistical physics. 2005 Nov 1;121(3-4):343-60.

Jacquet P, Seroussi G, Szpankowski W. On the entropy of a hidden Markov process. Theoretical computer science. 2008 May 1;395(2-3):203-19.

Minor comments

=====

- (Line 62) A new paper has been published on unsupervised learning of USVs. Maybe you can consider including it: <https://arxiv.org/pdf/2003.05897.pdf> Note that this paper deviates from VoiCE/MUPET since newer clustering techniques are used, showing advantages over k-means.

We thank the reviewer for pointing out this paper. We have added a citation to this paper in the introduction.

- (Line 107) When fitting a distribution over data, you should not use a R^2 statistic for the goodness-of-fit, but rather a test for fitting distributions. Wikipedia has a nice list here: https://en.wikipedia.org/wiki/Goodness_of_fit#Fit_of_distributions
- (Line 116) Same as above, use a different test.

We thank the reviewer for these comments. We have applied a K-S test and updated the manuscript accordingly.

- (Line 168) Remove the word "large" (what is a large deviation vs. a deviation?)

removed.

- (Line 289) What does a semi-colon ; mean in this notation?

We have followed the standard notation in information theory where the mutual information between two random variables X and Y is denoted as $I(X; Y)$ (see for example pages 19 and 24 in Cover and Thomas, 2005). In the case of line 289, the first random variable is X_n and the second is X_{n-1}, \dots, X_{n-D} .

- (Line 340) Could be useful mention that this is the definition of KL divergence (for completeness).

This is now mentioned in the revised manuscript.

- (Line 349) The use of "expected" seems inappropriate: What (or why) is expected?

We have explained better what we mean by "expected". The new text reads: "The pairs of Simple-long and Simple-short appear more than expected from their occurrence probabilities (figure 2B) assuming independence."

- (Line 387) The problem with the clustering algorithms not only lies on the lack of separability between clusters, but rather that this problem does not have a natural measure of similarity between USVs. This seems to be the major difficulty to find good representations through unsupervised learning.

We have correspondingly changed the relevant text (lines 409-414).

- (Line 409) bonded -> bounded?

Yes, corrected.

- (Line 429) extremely -> low. Extreme seems arbitrary.

We removed the word extremely.

- Figure 7: Both plots in A should have the same scale in the y-axis, so that they can be more easily compared.

We thank the reviewer for this comment. We have updated the y-axis scale in fig. 7A of the revised manuscript such that both plots now have the same scale.

- Paragraph starting at 445: The authors mention only backward-dependency in time. Seems reasonable that there could be backward and forward dependency.

While this is a reasonable possibility, in this study we have chosen to take a Markovian point of view which is typically stated in terms of backward time, and under this assumption, the suffix holds all the relevant information.

- (Line 467) This problem does not necessarily have a "true" classification. Moreover, a wider set of features might not be needed, but a better measure of similarity between USVs to discriminate between them.

We improved the following text to make these points clearer.

- (Line 472) True separation is not technically correct. USVs clusters are either separable or not. We improved the related text to make this clearer.

- (Line 497) I do not agree with the the "unexpected" motifs comment. The occurrence probabilities can be very noisy due to biases introduced in the clustering algorithms, which is already known by the authors: the different algorithms (MSA, VoiCE, MUPET) do cluster different USVs very differently. Knowing this, what is expected? The expected probabilities depend on the bias introduced by each algorithm.

We refer here to the expectation post labeling and added a note about that in the relevant text. Basically, in the specific literature of USVs modeling, motifs are often referred to as repetitions of specific short sub-sequences. For example, ABC within the sequence ... **ABC**BBCAB**ABC** ... without considering the probability of this event occurring. In the relevant text, we try to emphasize that such events are mainly of interest if their occurrence deviates from the expected probability given the null hypothesis of independence occurrence of USVs. Indeed, the exact same vocalization signal is labeled differently by different algorithms, therefore very different motifs can appear for each of the labeled data. We refer here to the expectation post labeling. In other words, given a labeled sequence, what is the probability for a given motif to occur. For example, if an algorithm would assign all the syllables with the same label A independent of their spectral features, then the motif AAA will be highly abundant, but it would be very "uninteresting" because it is completely predictable.

- (Line 512) I think that the word "demonstrated" is too strong in this setting, which is empirical. I would change it for "suggests".

We have updated the corresponding line..

- (Line 514) After "original one", a new paragraph should start. In that line, you start discussing the computational complexity of your implementation, which is different than the previous idea and might create confusion for readers that are less computationally-inclined.

Done, thank you.

- (Lines 543-548) The authors should also mention the trainability of increasing the number of classes. More data is needed to not overfit. Moreover, you use maximum likelihood (line 685) to estimate the transition probabilities. It would be a good addition to include how much data is needed.

We have added text (lines 594-598) discussing the effect that increasing N_c has on the required amount of data.

- (Line 689) Either use X_i or x_i for random variables, but not both. Changing the notation might be confusing for some readers.

We have clarified in the text that X_i represents the random variable and x_i represents the realization (observed value) of the variable. We have double-checked that this is consistent throughout the manuscript.

- (Definitions and theorems) Remove the equation numbers coming from the reference from which you obtained them, as this is not part of the format of your paper.

Thanks, we have removed the equation numbers.

- (Line 703) Add "Y =" before $X_{\{n-d\}}$...

Corrected.

- (Line 705) which is the stored -> which is stored

Thanks, we updated the text accordingly.

- (Line 707) Add that you use total probability to compute the joint density.

We have mentioned the law of total probability in the revised manuscript.

- (Line 725) A note about how MI measures independence of random variables would be useful.

We have added the following line: "Therefore, if X and Y are statistically independent then the mutual information $I(X; Y)$ is 0."

- (Line 751) Use a notation for vectors such as underbar or bold to make clear that you are dividing entry-wise, otherwise it is a bit hard to read.

Thanks, we have added an underbar.

Final remarks - A thought experiment

=====

Let us assume we have a Markovian sequence of order p . That sequence is corrupted by a structured noise (could be Markovian noise) which changes the labels of USVs with some probability. This could potentially create new motifs not originally existing in our sequence.

Let us assume that we have an estimate of the cluster to which each symbol belongs to (i.e. your initialization of the SIS) obtained from the noisy version of the sequence, and that you run SIS and find motifs.

What have you learned?

With this thought experiment, I am trying to point out that it is not straightforward to claim what has one "learned" (in the statistical learning sense) with such an approach. When the unsupervised learning algorithms give a biased sequence, one might learn the predictability of the next symbol of the "true signal" (if it exists) plus noise (introduced by the bias of the initial selection of clusters). A note on this topic in the discussion would be appreciated. Moreover, the authors could include that this is partly a limitation of using Markov Models (instead of, for example, Hidden Markov Models), since the observation is assumed to be the actual label.

We thank the reviewer for raising these thoughtful points. We have expanded the discussion (lines 510-519) to include the difficulty imposed by the bias of the clustering algorithms. Indeed, the geometry of the clusters may impose biases, and this could create a false structure in the sequence.

Note, however, that within the context of current clustering algorithms, the algorithm considers each syllable individually, therefore it has no information about the past or future and thus its bias cannot be Markovian in any way. One can feed the algorithm the syllables in any order and receive the exact same labeling every time.

It is, therefore, the case that the bias is "true noise" (i.e. is independent of the signal, which in our case is the structure of the sequence) and because of that, it may decrease our chance in predicting the next syllable but it cannot improve it (in the SIS setup). Therefore, if there is a structure in the temporal order of the USVs, the biases introduced by the different algorithms reduce the MI between the suffix and the next syllable. If we had the right clustering, then this type of noise would be minimized and the MI between the suffix and the next syllable would be maximized. As mentioned in the answer to the reviewer's second comment, we agree about the note regarding HMM and added text about it in the discussion.

REVIEWERS' COMMENTS:

Reviewer #5 (Remarks to the Author):

I appreciate the thoughtfulness included in the answers to my questions/concerns. I am satisfied with the responses.

I have two small comments:

- The reference of the paper I suggested is missing the arXiv link or pre-print number.
- I can't see the changes to Figure 7, as the rebuttal document suggests. When reviewing this, I noticed that the same original comment applies to Figures 6B & 6C: The y-axis on these should be the same, for easier comparison of the information provided in different plots.